# Sub-centrosomal mapping identifies augmin-γTuRC as part of a centriole-stabilizing scaffold

Nina Schweizer [1], Laurence Haren [2], Ilaria Dutto[1], Ricardo Viais [1], Cristina Lacasa[1], Andreas Merdes [2] & Jens Lüders [1]✉

Centriole biogenesis and maintenance are crucial for cells to generate cilia and assemble centrosomes that function as microtubule organizing centers (MTOCs). Centriole biogenesis and MTOC function both require the microtubule nucleator γ-tubulin ring complex (γTuRC). It is widely accepted that γTuRC nucleates microtubules from the pericentriolar material that is associated with the proximal part of centrioles. However, γTuRC also localizes more distally and in the centriole lumen, but the significance of these findings is unclear. Here we identify spatially and functionally distinct subpopulations of centrosomal γTuRC. Luminal localization is mediated by augmin, which is linked to the centriole inner scaffold through POC5. Disruption of luminal localization impairs centriole integrity and interferes with cilium assembly. Defective ciliogenesis is also observed in γTuRC mutant fibroblasts from a patient suffering from microcephaly with chorioretinopathy. These results identify a non-canonical role of augmin-γTuRC in the centriole lumen that is linked to human disease.

[1] Mechanisms of Disease Programme, Institute for Research in Biomedicine (IRB Barcelona), The Barcelona Institute of Science and Technology (BIST), 08028 Barcelona, Spain. [2] Molecular, Cellular and Developmental Biology, Centre de Biologie Intégrative, CNRS-Université Toulouse III, 31062 Toulouse, France. ✉email: jens.luders@irbbarcelona.org

Centrioles, which are at the core of the centrosome and template the formation of cilia, are formed by nine sets of microtubules that are arranged in a circular fashion so that they form the wall of a cylinder. In human cells, wall microtubules of mature centrioles are organized as triplets and doublets in the proximal and distal cylinder, respectively. Triplets consist of one complete microtubule, the so-called A-tubule, and two incomplete B- and C-tubules that share part of their wall with the adjacent A- and B-tubule, respectively. Doublets consist of only A- and B-tubules[1–3]. In cycling cells formation of new centrioles is initiated during S phase and occurs laterally at mother centrioles[4]. The origin of centriolar wall microtubules is not clear. The finding that the microtubule nucleator γTuRC is required for centriole biogenesis[5–11] and the observation that conical structures cap the minus-end of A-tubules[12], suggest that at least the A-tubules arise by γTuRC-mediated nucleation. During S/G2 phase daughter centrioles elongate and, after passing through mitosis, are converted to centrosomes through acquisition of PCM[13]. The PCM is the canonical site of γTuRC localization, where it has a well-established role as nucleator of microtubules that extend into the cytoplasm during interphase and are incorporated into the spindle during mitosis[14–16]. The cone-shaped γTuRC is assembled from γ-tubulin complex proteins (GCPs) 2–6, MZT1, MZT2, an actin-like protein, and multiple γ-tubulin molecules at its open face that have been proposed to function as template for α-β-tubulin assembly[17–21]. Centrosome targeting of γTuRC is regulated by accessory factors such as NEDD1[5,22]. Electron microscopy (EM) and, more recently, super-resolution microscopy have revealed localization of γTuRC subunits also at the subdistal appendages[23–27] and in the lumen of centrioles[6,28–34], but their roles at these sites were not investigated. During ciliogenesis, the mother centriole transforms into a basal body and templates formation of the axoneme, a microtubule-based structure that is at the core of cilia. However, axoneme microtubules are believed to not require nucleation but originate from elongation of the doublet microtubules in the distal basal body wall[35].

Here we have re-evaluated the long-standing view that γTuRC is a component of the PCM and that its centrosomal role is to nucleate microtubules. We found that γTuRC is distributed as functionally distinct subpopulations on the outside and, in complex with augmin, in the lumen of centrioles. Luminal augmin-γTuRC does not nucleate microtubules but contributes to centriole integrity, maintaining the ability of centrioles to template cilium formation.

## Results

**γTuRC forms separable centrosomal subpopulations**. To identify potential centrosomal subpopulations of γTuRC and elucidate whether these may have distinct functions, we analyzed the centrosomal localization of the γTuRC targeting factor NEDD1[5,22] and of the core subunits γ-tubulin and GCP4 by expansion microscopy (ExM)[36]. γTuRC subunits localized on the outer surface of both mother and daughter centrioles, visualized with anti-acetylated α-tubulin antibodies, in some cases displaying enrichment in the proximal part of mother centrioles. In addition, all three proteins were found in the centriole lumen (Fig. 1a, c, d, e). This localization pattern was fundamentally different from that of the bona fide PCM components CDK5RAP2 and pericentrin, which associated only with the outer, proximal part of mother centrioles (Fig. 1a). To re-evaluate the paradigm that centrosomal microtubules are nucleated in the PCM, we analyzed microtubule regrowth after cold-induced depolymerization. Whereas microtubules could not be detected in cold-treated cells, after a few seconds of warming microtubules were nucleated in close proximity of centriole cylinders (Fig. 1b).

Microtubules grew preferentially from the proximal surface of mother centrioles, but were also observed along the entire centriole wall including at distal ends. Thus, γTuRC and microtubule nucleation activity are generally associated with the outer surface of centrioles, with some enrichment in the PCM.

**γTuRC colocalizes with augmin in the central lumen of centrioles**. Next, we focused on luminal γTuRC. Interestingly, we found that HAUS6, a subunit of the augmin complex, which recruits γTuRC to spindle microtubules during mitosis[37,38], colocalized with γTuRC in the centriole lumen during interphase (Fig. 1c). Luminal localization was observed for both endogenous augmin subunits and EGFP-tagged recombinant versions (Fig. 1c, Supplementary Fig. 1a). By comparing the localizations of HAUS6 and NEDD1 relative to SAS-6, a component of the cartwheel structure in the proximal lumen of daughter centrioles[39], and centrin, a marker for the distal lumen of centrioles[40], we found that in all cases NEDD1 and HAUS6 were found distal to the SAS-6 signal (Fig. 1d) and proximal to the bulk of the centrin signal (Fig. 1e). These results show that apart from γTuRC that is recruited to the outside of centrioles, a separate pool colocalizes with augmin in the central region of the centriole lumen.

**Luminal recruitment of augmin and γTuRC occurs late during centriole elongation**. Interestingly, newly formed daughter centrioles frequently lacked luminal augmin and γTuRC, even though γTuRC was visible on the outside of the daughter cylinder (Fig. 1e). Consistent with this, only a minor fraction of all daughter centrioles identified by centrin labeling, were also positive for luminal HAUS6 or NEDD1 (Fig. 1f). Measuring the distance between centrin foci of mother and daughter centrioles as a proxy for daughter centriole length revealed that these centrioles were more elongated than daughter centrioles that lacked these proteins (Fig. 1g), suggesting that augmin and γTuRC localize to the lumen late during centriole biogenesis. Corroborating this result, accumulation of HAUS6 in the lumen of daughters coincided with poly-glutamylation, a tubulin modification that occurs selectively on the C-tubules of triplet microtubules[41] and that became detectable only after daughters had reached a substantial length (Supplementary Fig. 1b). The most robust luminal HAUS6 signal was observed in mature centrioles, where it was confined to the proximal/central cylinder, marked by poly-glutamylation (Supplementary Fig. 1b).

Together our results indicate that γTuRC localizes first to the outside of newly formed daughter centrioles and subsequently, during centriole elongation, accumulates with augmin in the centriole lumen (Fig. 1h).

**Centriole outer wall recruitment of γTuRC depends on CEP192**. Previous work implicated CEP192 in the recruitment of γTuRC to centrosomes[42,43] and identified the targeting factor NEDD1 as proximity interactor of CEP192[44,45], but did not distinguish between the distinct subcentrosomal sites. Super-resolution microscopy detected CEP192 along the outer side of mother and daughter centrioles[46,47]. To re-evaluate CEP192's role in γTuRC centrosome recruitment, cells were transfected with siRNA, synchronized in mitosis with the Eg5 inhibitor STLC and then, bypassing cell division, released into G1 by CDK1 inhibition (Fig. 2a). This setup avoided adverse effects on centrosomes from duplication failure, and enriched interphase cells with fully elongated, CEP192-depleted centrioles (Fig. 2b). Using ExM and ultrastructure expansion microscopy (U-ExM)[41], we found that CEP192 depletion removed both NEDD1 and γ-tubulin specifically from the outside of centrioles, whereas the luminal pool of

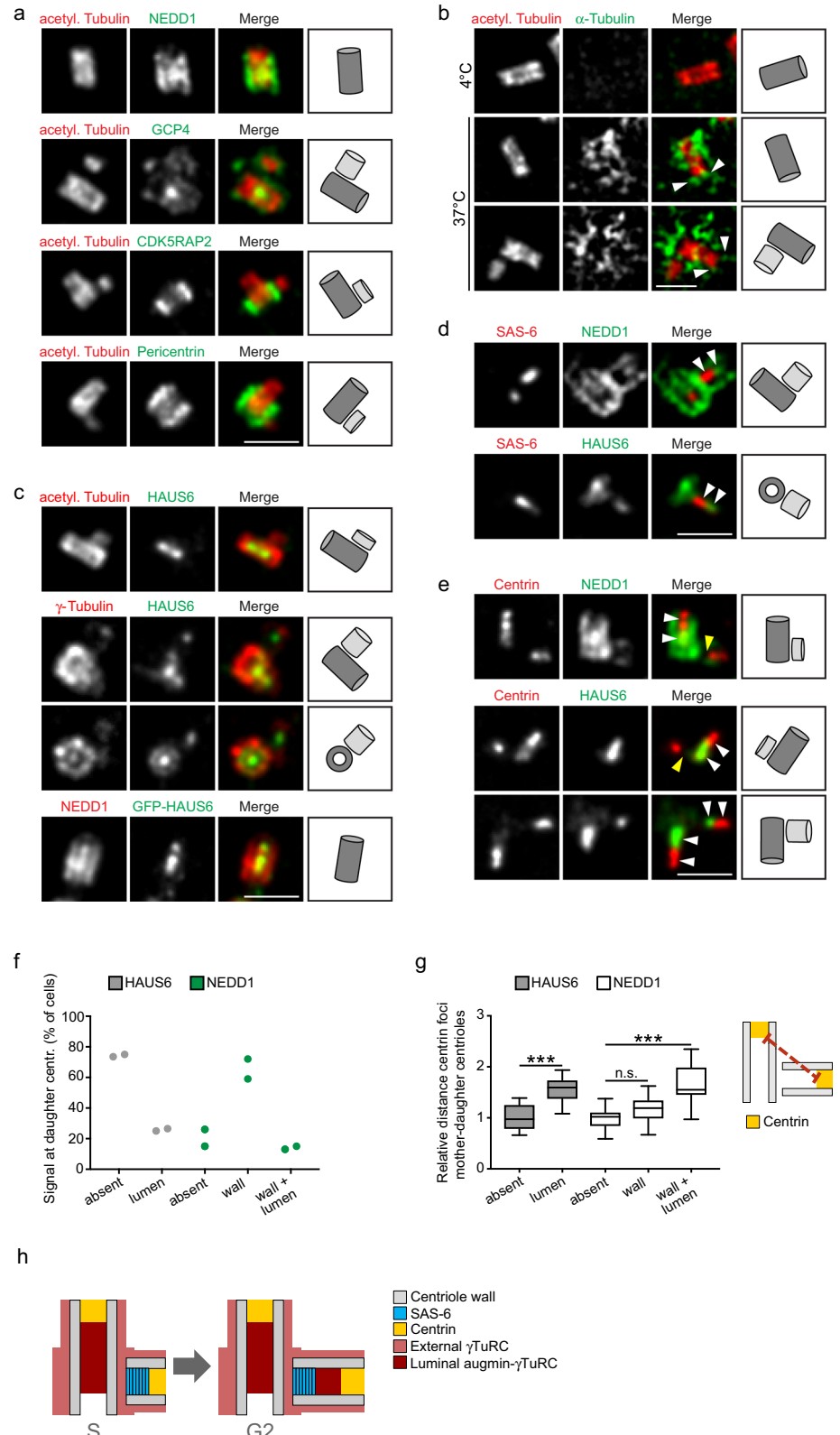

these proteins appeared unaffected (Fig. 2c and Supplementary Fig. 2a, b). Luminal HAUS6 signal was also unaffected in these cells (Supplementary Fig. 2b). Despite very efficient depletion of CEP192 and in line with specific removal of γTuRC from the outside of centrioles, but not from the lumen, overall centrosomal γ-tubulin levels were reduced by ~50% (Fig. 2h).

**Luminal recruitment of γTuRC requires augmin**. Considering their colocalization (Fig. 1c), we asked whether luminal recruitment of γTuRC required augmin. Using the same experimental setup as before, we depleted cells of HAUS6 (Supplementary Fig. 2c). Strikingly, ExM revealed that centrioles lacking HAUS6 also lacked γ-tubulin in the lumen (Fig. 2d), whereas γ-tubulin

**Fig. 1 γTuRC forms distinct centrosomal subpopulations. a** Centrioles of U2OS cells in ExM stained for acetylated α-tubulin and either NEDD1, GCP4, CDK5RAP2, or pericentrin. **b** Centrioles and microtubules of interphase U2OS cells in ExM stained for acetylated α-tubulin and α-tubulin in a microtubule regrowth assay. Depicted is the condition before (4 °C) and after (37 °C) microtubule regrowth. Arrowheads point to microtubules associated with the distal centriole wall. **c** Centrioles of parental U2OS cells or U2OS cells stably expressing EGFP-HAUS6 in ExM stained for acetylated α-tubulin and HAUS6, γ-tubulin and HAUS6 or NEDD1 and GFP (EGFP-HAUS6). **d** Centrioles of U2OS cells in ExM stained for SAS-6 and either NEDD1 or HAUS6. Arrowheads point to adjacent signals of NEDD1/SAS-6 or HAUS6/SAS-6 in the daughter centriole lumen. **e** Centrioles of U2OS cells in ExM stained for centrin and either NEDD1 or HAUS6. White arrowheads point to adjacent signals of NEDD1/centrin or HAUS6/centrin in the centriole lumen. Yellow arrowheads point to daughter centrioles that lack NEDD1 or HAUS6 in the lumen. **f** Quantifications of the percentage of cells with HAUS6 or NEDD1 at the wall/lumen of daughter centrioles from 2 independent experiments, 100–102 cells per condition and experiment. **g** Quantifications of the relative distance between the centrin foci of mother and daughter centrioles in cells where HAUS6 or NEDD1 are absent/present at daughter centrioles. Boxes contain pooled data from 5 (HAUS6) or 3 (NEDD1) independent experiments, 3–16 mother-daughter centriole pairs analyzed per condition and experiment ($p$ (HAUS6 absent/lumen) <2.22e-16, $p$ (NEDD1 absent/wall) = 0.131, $p$ (NEDD1 absent/wall and lumen) = 1.6653e-15, mixed-effects linear model). Boxes extend from the 25th to the 75th percentile, the horizontal line indicates the median, whiskers depict the 10th and 90th percentiles. **h** Cartoon summarizing the localizations of distinct γTuRC subpopulations. Cartoons in (**a–e**) illustrate configurations of mother (dark grey) and daughter (light grey) centrioles. Scale bars (all panels), 2 μm. Acetyl. Tubulin, acetylated α-tubulin; centr., centrioles. ***$p$ < 0.001; n.s., not significant. Source data are provided as a Source Data file.

signals on the outside of centrioles could still be detected (Fig. 2d). Similar to CEP192 depletion, this correlated with a partial decrease of overall centrosomal γ-tubulin levels (Fig. 2i). Importantly, we never observed HAUS6-negative centrioles that were positive for luminal γ-tubulin. To test if luminal localization of γ-tubulin required only HAUS6 or also other augmin subunits, we depleted cells of HAUS5. Under this condition we detected centrioles that lacked both luminal γ-tubulin and HAUS6 (Fig. 2e), consistent with a requirement of the augmin complex rather than HAUS6 alone for luminal localization.

Since previous work showed that γTuRC centrosome targeting in cycling cells is broadly mediated by NEDD1[5,22], we also tested depletion of NEDD1. In this condition NEDD1 and γ-tubulin were absent from both the outside and the lumen of centrioles (Fig. 2f, g), whereas HAUS6 localization was unaffected (Supplementary Fig. 2d). Thus, outer wall localization of γTuRC requires CEP192, luminal γTuRC localization depends on augmin, and the targeting factor NEDD1 is required for recruitment to both sites.

We additionally analyzed γTuRC lumen localization in mouse hippocampal neurons in which *Haus6* had been knocked out post-mitotically. While HAUS6 was still detectable at centrioles in conditional *Haus6* KO neurons at two days in vitro (DIV) (Supplementary Fig. 2e), several days later, at nine DIV, it was largely absent from this site (Supplementary Fig. 2f, g). We previously showed that during neuronal culture PCM-associated γ-tubulin is strongly downregulated and the remaining signal is centriole-associated[48]. Indeed, in neurons at nine DIV γ-tubulin displayed centriolar localization (Supplementary Fig. 2f). Strikingly, after conditional *Haus6* KO centriolar γ-tubulin was hardly detectable (Supplementary Fig. 2f, g). These findings indicate that residual centrosomal γTuRC in differentiated neurons is luminal and recruited by the same augmin-dependent mechanism as in cycling cells.

**Augmin interacts with the inner scaffold protein POC5.** To learn more about the roles of augmin and γTuRC in the centriole lumen, we performed biotin proximity labeling using HAUS6 fused to the BirA biotin ligase as bait (Fig. 3a). In fixed cells biotinylated proteins were detected in the mitotic spindle and at centrosomes (Supplementary Fig. 3a), sites at which augmin is known to localize. Streptavidin affinity capture of biotinylated proteins coupled to mass spectrometry identified all augmin subunits and the γTuRC subunit GCP3, which served to validate our approach (Fig. 3a; Supplementary Data 1). In addition, we identified POC5, a centrin-binding protein and component of a scaffold structure at the luminal surface of centrioles proposed to protect against mechanical

stress[49–52]. Consistent with this, augmin subunits were previously found as proximity interactors in cells expressing POC5-BirA as bait[44]. POC5 was present in the lumen of mother centrioles and a fraction of daughter centrioles (Fig. 3b), resembling the behavior that we had observed for luminal augmin and γTuRC. POC5 and γ-tubulin were confined to the same central luminal region, but in end-on views γ-tubulin appeared to localize more interior than POC5 (Fig. 3b). To more precisely map the relative localizations of POC5, augmin, and γTuRC in the centriole lumen, we used U-ExM in combination with super-resolved imaging by random illumination microscopy (RIM)[53]. This revealed that POC5 localized closest to the centriole wall, marked by acetylated α-tubulin staining (Fig. 3c). γ-Tubulin was labeled poorly on the outside of centrioles by U-ExM, but was robustly labeled in the lumen. We found that γ-tubulin localized more luminal than POC5, not in direct contact with the wall (Fig. 3c). HAUS6 appeared to localize in between the regions occupied by POC5 and γ-tubulin (Fig. 3c).

**POC5 is required for luminal recruitment of augmin-γTuRC.** Given this specific localization pattern and since we occasionally observed daughter centrioles that were positive for POC5, but lacked γ-tubulin in the lumen (Fig. 3b), we tested whether POC5 may function upstream of augmin and γTuRC lumen recruitment. Using the mitotic arrest-release approach in control and POC5 RNAi cells (Supplementary Fig. 3b), we found that centrioles depleted of POC5 also lacked luminal HAUS6 and γ-tubulin, whereas both proteins were always present at centrioles in control cells (Fig. 3d). Previous analysis by cryo-electron tomography (cryo-ET) showed that the inner scaffold is a periodic, helical structure, lining the inner centriole wall[51], likely composed of repeating units of scaffold protein complexes. Thus, scaffold proteins may tend to self-associate. Indeed, POC5 exogenously expressed in human cells forms filamentous structures in the cytoplasm that associate with other centriole proteins[49]. We confirmed this observation and found that these ectopic assemblies were also positive for HAUS6 and γ-tubulin (Fig. 3e, Supplementary Fig. 3c). Together, these findings demonstrate that POC5, through interaction with augmin, recruits γTuRC to the inner centriole scaffold.

**POC5, augmin, and γTuRC promote centriole integrity.** The inner scaffold was shown to confer stability on centrioles[51,52]. When we quantified the number of centrioles at the end of the duplication cycle, by counting centrin foci in mitotic cells, we found that POC5 depletion had no effect on the number of centrioles (Supplementary Fig. 4a). We also did not observe any change in centriole number after HAUS6 depletion

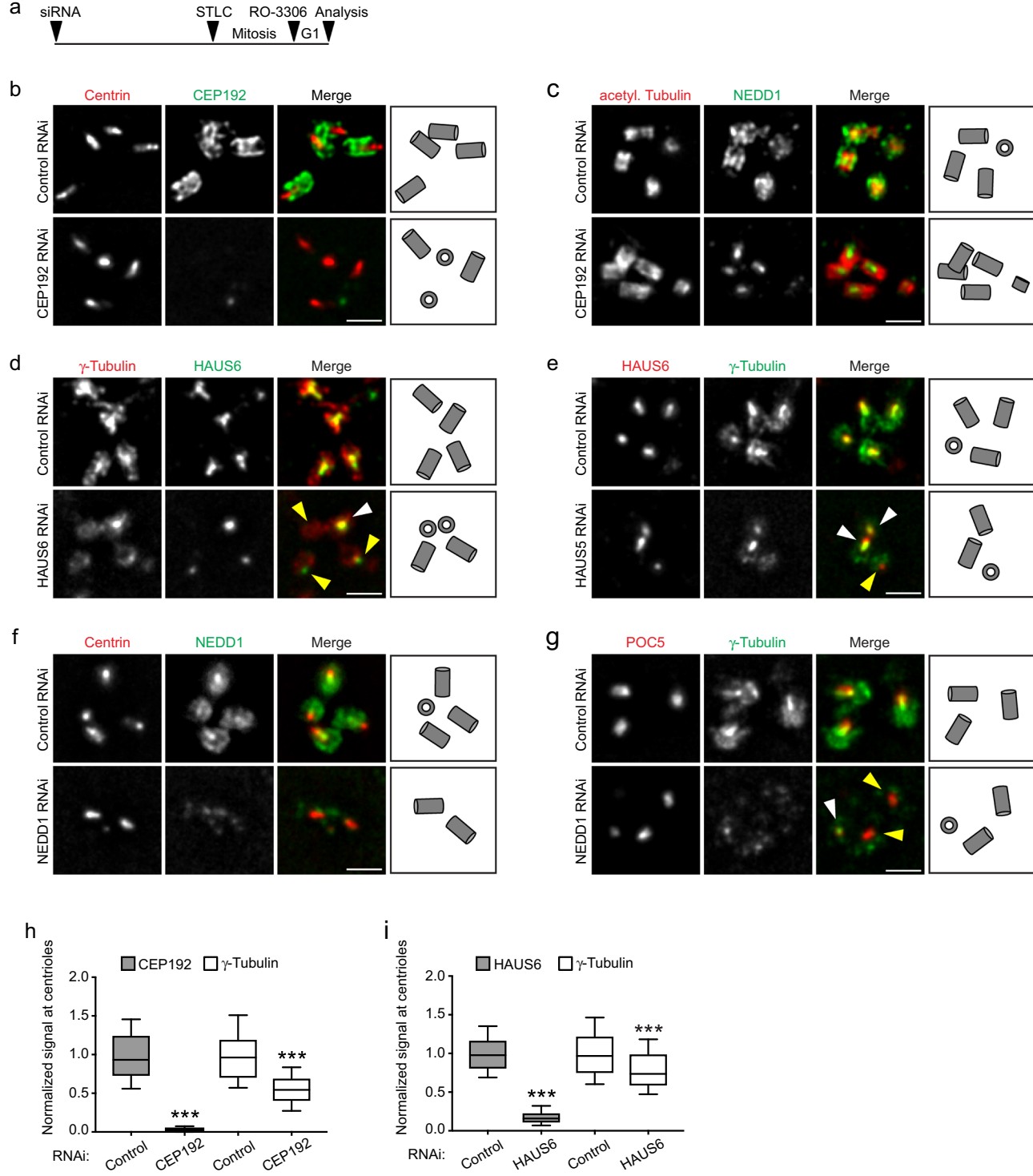

(Supplementary Fig. 4b), confirming earlier reports[37]. However, when mitotic duration was extended up to ~18 hours by treatment with the Eg5 inhibitor STLC, centriole numbers in POC5 and HAUS6 RNAi cells declined, whereas 70-80% of control cells still had the expected number of at least four centrin foci, this number was observed in only ~35% of POC5-depleted cells and ~50% of HAUS6-depleted cells (Fig. 4a–d), suggesting centriole destabilization. Time-course experiments further revealed that the decline in centriole numbers correlated with the time cells spent in mitosis (Fig. 4e). We also assayed centriole stability after HAUS6 depletion in noncancerous RPE1 cells. To avoid p53-

dependent G1 arrest caused by mitotic defects[54], we induced cell cycle exit by serum withdrawal immediately after transfection of siRNA (Supplementary Fig. 4c). After HAUS6 was efficiently depleted, we added serum for cell cycle re-entry, and STLC for prolonged mitotic arrest. Similar to U2OS cells, albeit less pronounced, centrioles in HAUS6-depleted RPE1 cells were also destabilized (Supplementary Fig. 4c). γTuRC is required for centriole assembly, possibly by nucleating microtubules of the centriole wall[5–8,12]. To determine if γTuRC additionally participates in centriole stabilization following centriole assembly, we compared centriole numbers in control and GCP4-depleted

**Fig. 2 γTuRC centriole localization depends on CEP192 and augmin. a** Schematic depicting the experimental design in (**b–i**). **b, c** Centrioles of control RNAi and CEP192 RNAi U2OS cells in ExM stained for centrin and CEP192 (**b**) or acetylated α-tubulin and NEDD1 (**c**). **d** Centrioles of control RNAi and HAUS6 RNAi U2OS cells in ExM stained for γ-tubulin and HAUS6. White arrowhead points to a centriole that is not depleted of γ-tubulin, yellow arrowheads point to centrioles that lack HAUS6/luminal γ-tubulin. **e** Centrioles of control RNAi and HAUS5 RNAi U2OS cells in ExM stained for HAUS6 and γ-tubulin. White arrowheads point to centrioles that are not depleted of HAUS6 and γ-tubulin, yellow arrowhead points to a centriole that lacks HAUS6/luminal γ-tubulin. **f, g** Centrioles of control RNAi and NEDD1 RNAi U2OS cells in ExM stained for centrin and NEDD1 (**f**) or POC5 and γ-tubulin (**g**). White arrowhead points to a centriole that has γ-tubulin in the lumen, yellow arrowheads point to centrioles that are depleted of γ-tubulin at the wall and in the lumen. Scale bars (all panels), 2 μm. Cartoons illustrate centriole configurations. **h** Relative fluorescence signals for CEP192 and γ-tubulin at centrioles in control RNAi and CEP192 RNAi U2OS cells. The box plot contains pooled data from three independent experiments, 19–55 centrioles analyzed per condition and experiment ($p$ (CEP192) = 2.3595e-05, $p$ (γ-tubulin) = 0.00069325, mixed-effects linear model). **i** Relative fluorescence signals for HAUS6 and γ-tubulin at centrioles in control RNAi and HAUS6 RNAi U2OS cells. The box plot contains pooled data from three independent experiments, 51–60 centrioles analyzed per condition and experiment ($p$ (HAUS6) = 1.0088e-05; $p$ (γ-tubulin) = 8.2174e-05, mixed-effects linear model). Boxes in (**h**) and (**i**) extend from the 25th to the 75th percentile, the horizontal line indicates the median, whiskers depict the 10th and 90th percentiles. Acetyl. Tubulin, acetylated α-tubulin. \*\*\*$p$ < 0.001. Source data are provided as a Source Data file.

U2OS cells that had spent different times in mitosis using the above synchronization protocol (Fig. 4f, g). At 30 minutes in mitosis, there was no difference in centriole numbers between control and GCP4-depleted cells (Fig. 4g), implying that centriole duplication was not affected under our experimental conditions. However, counting centrioles in cells that had spent 24 h in mitosis revealed a clear reduction of centriole numbers after GCP4 depletion (Fig. 4f, g), suggesting that γTuRC, like POC5 and augmin, is required for centriole stability.

Curiously, during prolonged mitotic arrest the number of centrioles in control cells also decreased (Fig. 4e, g). We speculated that this was due to premature loss of the cartwheel, which normally occurs at mitotic exit in a PLK1-dependent manner[13,55]. Indeed, in the presence of the PLK1 inhibitor BI2536 the cartwheel component SAS-6 was retained at daughter centrioles during prolonged mitotic arrest (Supplementary Fig. 4d, e) and centriole destabilization after HAUS6 depletion was partially rescued (Fig. 4e). Thus, luminal augmin may be particularly important for the stability of cartwheel-less daughter centrioles.

The time-dependent disappearance of centrin foci in mitotically arrested cells may indicate complete centriole disassembly or merely loss of their distal, centrin-containing compartment. To distinguish between these possibilities, we quantified foci of centrobin, which localizes to the outer wall of daughter centrioles, in a central region[56,57]. In contrast to the reduction in centrin foci, a similar percentage of control and HAUS6-depleted cells had the expected number of at least two centrobin foci (~88% vs. ~83%) (Fig. 4h, i), demonstrating that the majority of centrioles did not completely disassemble.

**Augmin promotes proper centriole length.** Augmin depletion induces mitotic delay[37,38,58,59], which per se could interfere with centriole architecture. For example, it was recently shown that centrioles over-elongate during prolonged mitosis in a PLK1-dependent manner[60]. To determine if augmin-depleted centrioles show structural abnormalities, independent of prolonged mitosis, we treated control and HAUS6-depleted U2OS cells with the Aurora kinase inhibitor VX-680, to override the spindle assembly checkpoint, and analyzed centrioles by U-ExM. In the presence of VX-680 cells are able to enter mitosis, but exit shortly after without cell division, resulting in polyploidy[61,62]. Indeed, incubation with VX-680 for three days resulted in enlarged cells that contained extra centrioles both in control and HAUS6 depletion conditions (Fig. 5a), consistent with continued cell cycle progression. Whereas centrioles in VX-680-treated control cells contained luminal HAUS6, this signal was strongly reduced in centrioles of VX-680-treated cells after HAUS6 RNAi (Fig. 5a). This was evident also in mature centrioles, identified by the presence of the subdistal appendage marker ninein (Fig. 5a).

Strikingly, while the diameter of HAUS6-depleted centrioles was normal, they were significantly shorter than control centrioles (Fig. 5b–d). This difference in centriole length was also observed when considering only mother centrioles in S or G2 phase, identified by the presence of engaged procentrioles at their proximal end (Fig. 5b, d). Notably, in addition to the global reduction in centriole length (the distribution of lengths peaked between 450-500 nm for control centrioles and 400–450 nm for depleted centrioles), we also observed the presence of very short centrioles after HAUS6 depletion (~10% of centrioles < 250 nm in S/G2 cells vs. ~0.9% in control cells). These centrioles were positive for CP110 (Fig. 5b), a protein that localizes to the distal tip of centrioles[7], and had engaged procentrioles growing from their lateral surface, despite their reduced size. To corroborate this finding, we used RPE1 *p53* KO cells to generate an inducible *HAUS6* KO line. After induction, western blotting confirmed reduced HAUS6 protein levels indicating successful *HAUS6* KO (Fig. 5e). Similar to U2OS cells, treatment with VX-680 resulted in cells with extra centrioles (Fig. 5f). In VX-680-treated *HAUS6* KO cells we frequently observed centriole 'stubs', which either lacked HAUS6 or displayed only a faint HAUS6 signal at their distal tip (Fig. 5f–h). We also noticed an increase in centrioles that appeared broken or incomplete (Fig. 5f–h). Abnormal centrioles were labeled with α-tubulin antibodies and were positive for CP110 (Fig. 5i, j), confirming their identity as short centriole cylinders. As in U2OS cells, they were able to generate procentrioles (Fig. 5i, j). In summary, removal of augmin from centrioles resulted in reduced centriole length and abnormal architecture that were not an indirect result of mitotic delay.

**POC5 and γTuRC are required for ciliogenesis.** We next turned our attention to cilia assembly, which relies on the elongation of microtubule doublets at the distal tip of mother centrioles and thus might be particular sensitive to centriolar defects caused by the absence of inner centriole scaffold components. We initially considered studying ciliogenesis in *Haus6* knockout neurons. However, after *Haus6* conditional knockout loss of luminal HAUS6 required up to nine days of in vitro culture (Supplementary Fig. 2e–g), whereas ciliogenesis occurs much earlier[63,64]. Moreover, *Haus6* conditional knockout in neural progenitors of the developing mouse brain is embryonic lethal[65]. RPE1 cells are frequently used for assaying ciliogenesis. However, in contrast to U2OS cells, we never observed HAUS6-depleted mature mother centrioles after *HAUS6* KO in RPE1 *p53* KO cells, possibly due to the toxicity of *HAUS6* KO. Therefore, we tested depletion of POC5. RPE1 cells were treated with control or POC5 siRNA and serum-starved to induce ciliogenesis. Whereas 67% of control cells had a cilium, only ~30% of POC5-depleted cells that lacked POC5 on both centrioles were ciliated (Fig. 6a, b). Since POC5

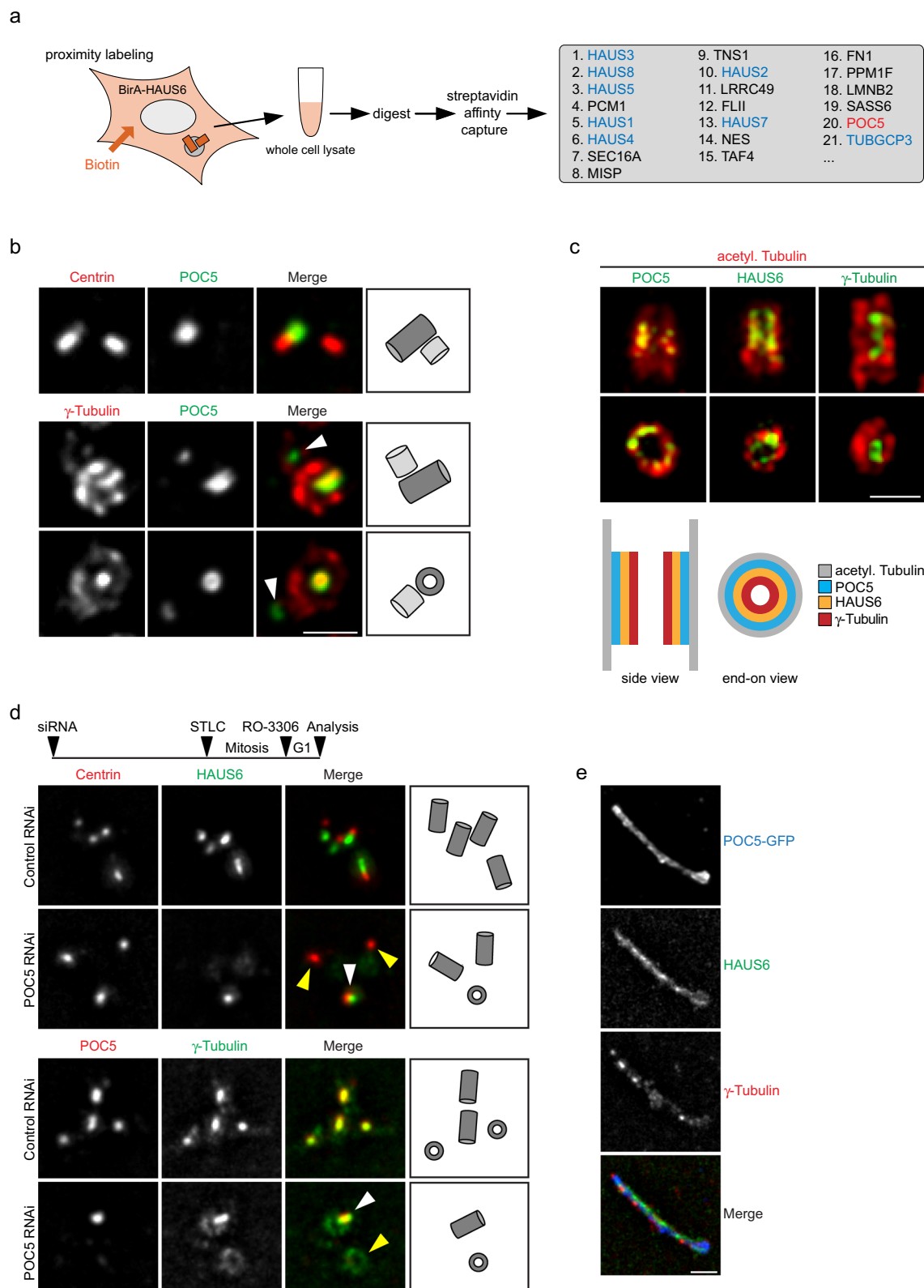

depletion was shown to cause G1 arrest in RPE1 cells[50], which could interfere with assaying ciliogenesis, we repeated the experiment in RPE1 *p53* KO cells. In this case ~55% of control cells were ciliated, whereas only ~14% of cells that lacked POC5 on both centrioles possessed a cilium (Fig. 6b), confirming that loss of POC5 from the centriole lumen impairs ciliogenesis.

Mutations in γTuRC subunits have been linked to developmental defects including primary microcephaly and retinopathy[66–70]. We hypothesized that some of the clinical manifestations may also involve centriole destabilization and impaired ciliogenesis. To address this, we analyzed cilium formation after serum starvation in GCP4 mutant fibroblasts, which were obtained from a patient

**Fig. 3 Augmin is recruited to the inner centriole scaffold by POC5. a** Mass spectrometry analysis of proximity interactors of BirA-HAUS6 from whole cell lysate of U2OS cells. The schematic depicts the experimental procedure, the box shows proteins identified by mass spectrometry and enriched in the BirA-HAUS6 sample relative to the BirA control sample. Proteins are ordered by spectral counts. The hit ACAB (Supplementary Data 1), a naturally biotinylated mitochondrial protein has been removed from the list as likely being background. **b** Centrioles of U2OS cells in ExM stained for centrin and POC5 or γ-tubulin and POC5. Arrowheads point to POC5 in the daughter centriole lumen. **c** Centrioles of U2OS cells in U-ExM stained for acetylated α-tubulin and either POC5, HAUS6, or γ-tubulin. Images were acquired with RIM. The cartoon depicts the relative localization of POC5, HAUS6, and γ-tubulin. **d** Centrioles of control RNAi and POC5 RNAi U2OS cells in ExM stained for centrin and HAUS6 or POC5 and γ-tubulin. White arrowheads point to centrioles with HAUS6 or γ-tubulin in the lumen, yellow arrowheads point to centrioles that lack HAUS6 or luminal γ-tubulin. The schematic depicts the experimental design. **e** POC5-GFP aggregate in ExM stained for GFP (POC5-GFP), HAUS6, and γ-tubulin. Scale bars (all panels), 2 μm. Cartoons in (**b,d**) illustrate mother (dark grey) and daughter (light grey) centriole configurations. Acetyl. Tubulin, acetylated α-tubulin.

diagnosed with microcephaly and chorioretinopathy and shown to contain a reduced amount of γTuRC[67]. Strikingly, only ~20% of GCP4 mutant fibroblasts were ciliated compared to ~80% of control fibroblasts (Fig. 6c, d). Additionally, cilia in patient fibroblasts were significantly shorter (Fig. 6e). Importantly, mother centrioles had acquired subdistal and distal appendages, as determined by the presence of ODF2, ninein, and CEP164, respectively (Fig. 6c, f), and were fully elongated (Fig. 6g), indicating proper maturation. To determine if GCP4 mutant cells had a reduced amount of γTuRC in the centriole lumen, we used U-ExM. Indeed, the luminal signal of GCP4 in patient cells was significantly reduced when compared to control cells, with the strongest effect in cells that were not able to mount a cilium (Fig. 6h). Similar ciliogenesis defects were also observed after RNAi-mediated GCP4 depletion in two different cells lines (Supplementary Fig. 5a–e), under conditions that did not impair centriole duplication (Supplementary Fig. 5a, d).

Together our data suggest that augmin and γTuRC, recruited by POC5 to the centriole lumen, are important for centriole integrity and ciliogenesis.

## Discussion

Here we have identified noncanonical roles of augmin and γTuRC in the centriole lumen that are independent of their previously described functions in microtubule nucleation. In addition to nucleating microtubules, both protein complexes are linked to the centriole inner scaffold through POC5, contributing to centriole integrity. POC5 was recently mapped to the innermost region of the scaffold, a position well suited for anchoring augmin-γTuRC[51,52]. Augmin is known to directly interact with microtubules through its HAUS8 subunit and this interaction is independent of its ability to recruit γTuRC[71–73]. Thus, it is tempting to speculate that augmin, apart from interacting with POC5, may also directly bind and stabilize microtubules of the centriole wall. Curiously, the shape and dimension of native and reconstituted augmin[71,72] have a striking similarity to unassigned Y- and L-shaped linker structures that are connected to A- and B-tubules of the centriole wall and protrude into the lumen[2,74,75]. Our analysis by U-ExM is consistent with such a configuration. The augmin subunit HAUS6 was found in an intermediate position relative to POC5, which is positioned closer to the inner centriole wall, and γ-tubulin, which is found more luminal. However, further work is needed to more precisely map the configuration of luminal augmin.

Considering the periodicity of the inner scaffold structure[51,74] and our observation that augmin and γTuRC are distributed along the entire length of the central lumen, one can speculate that multiple copies of augmin and γTuRC may adopt a stacked configuration at the inner centriole wall. Since both augmin and γTuRC are large multisubunit complexes, they would be expected to form prominent structures in the centriole lumen. Indeed, several EM and cryo-ET studies have described unidentified densities including ring-shaped structures in the centriole lumen[2,74,76,77]. The fact that these are not consistently observed may indicate sensitivity of augmin and γTuRC to the conditions used for sample preparation. While the precise arrangement of luminal augmin and γTuRC remains to be determined, we showed that their specific loss from the lumen impairs centriole integrity and ciliogenesis. Cilia assembly was also compromised after depletion of POC1B and WDR90[75,78], proteins that were found to assemble the inner scaffold at the luminal wall of centrioles[51,52]. More recently it was shown that in human cells depleted of WDR90 or POC5 centrioles display disrupted/incomplete walls and impaired roundness, defects that were exacerbated by co-depletion of both proteins[52]. We also observed structural centriole defects including shorter centrioles after HAUS6 depletion. Shorter centrioles were originally described also after POC5 depletion[50], but Steib et al. found that following mitosis and entry into G1 phase these elongated[52]. It should be noted that our analysis of centrioles after HAUS6 depletion required treatment of cells with Aurora kinase inhibitor, to allow depleted cells to pass through mitosis. Therefore, while the specific outcome of impaired centriole integrity may depend on the experimental setup, our finding that depletion of luminal augmin impairs centriole integrity is consistent with previous studies regarding the roles of inner scaffold proteins. Based on our results, augmin and γTuRC can be considered components of the inner scaffold, which may thus extend farther into the centriole lumen than previously anticipated. Our findings also suggest that the integrity of the entire extended scaffold structure is required for its centriole-stabilizing and ciliogenesis-promoting function.

While mother centrioles with wall defects may still allow growth of daughters from intact parts of their lateral surface, incomplete formation of the distal end, partial disassembly, or a compromised radial symmetry may interfere with the ability to template axoneme assembly. Indeed, we observed ciliogenesis defects in cells depleted of POC5 or GCP4, under conditions where centriole numbers were not altered.

Importantly, defective ciliogenesis was also observed in patient-derived γTuRC mutant fibroblasts. Thus, phenotypes previously ascribed to augmin or γTuRC deficiency in various human genetic disorders[66–68,70,79–81], may not be related solely to their function as microtubule nucleators, but also to their luminal role that permits ciliogenesis by ensuring centriole integrity.

## Methods

**Cell lines and culture**. U2OS cells, hTERT BJ, and hTERT RPE1 cells were cultured in DMEM and DMEM/F12 (hTERT RPE1) with 10% fetal bovine serum (FBS). Parental cell lines were obtained from ATCC, hTERT RPE1 *p53* KO cells were provided by Meng-Fu Bryan Tsou[82]. U2OS cells stably expressing POC5-GFP, EGFP-HAUS8, EGFP-HAUS6, or BirA-HAUS6 were generated by transfection of the appropriate expression plasmid using lipofectamine 2000 (Invitrogen), followed by either selection with 1 mg/ml geneticin or by FACS. Doxycycline-inducible *HAUS6* KO cells, or control cells inducible for Cas9, but lacking a sgRNA, were generated by lentiviral transduction of the appropriate plasmid into hTERT RPE1 *p53* KO cells. Clonal lines were established after single-cell sorting. Human fibroblasts were derived from skin biopsies from a control individual (WT) and a patient with GCP4 mutations (AII-1)[67] and cultured in DMEM with 15% FBS. All cell lines were kept in a 37 °C incubator with 5% $CO_2$ and a humidified atmosphere.

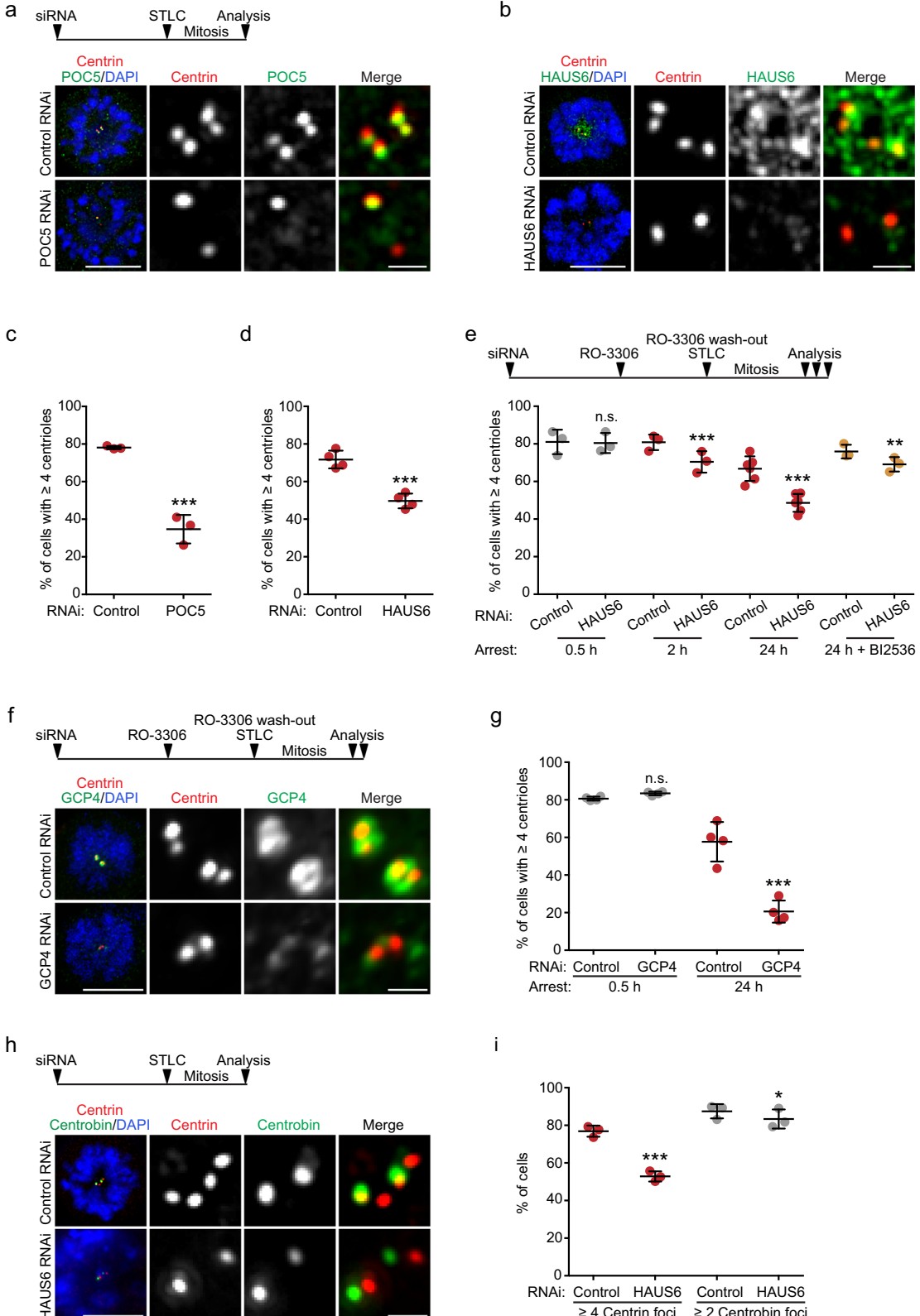

**RNA interference**. Depletion of CEP192, HAUS6, NEDD1, POC5, and GCP4 was performed by transfecting cells with the following siRNA oligonucleotides CEP192, 5′-AAGGAAGACAUUUUCAUCUCU-3′; HAUS6, 5′-CAGUUAAGCAGGUACG AA-3′; NEDD1, 5′-GCAGACAUGUGUCAAUUUGTT-3′; POC5, 5′-CAACAAA UUCUAGUCAUA-3′; GCP4 5′-GCTGCTTCATCAGATCAAT-3′; using Lipo-fectamine RNAiMAX (Invitrogen). siRNA oligos against luciferase (5′-UCGAA GUAUUCCGCGUACG-3′) were used as control.

**Cell culture treatments and assays**. Microtubule regrowth assay (Fig. 1b): cells grown on coverslips were incubated on ice for 40 min to depolymerize cytoplasmic microtubules. For microtubule regrowth, coverslips were transferred into 3.7% paraformaldehyde at 37 °C for 1 min, followed by the incubation in methanol at −20 °C for a minimum of 15 min. For the negative control (no microtubule regrowth), cells were incubated in 3.7% paraformaldehyde on ice for 10 min, fol-lowed by the incubation in methanol at −20 °C for a minimum of 15 min. Gen-eration of polyploid control or CEP192/HAUS6/HAUS5/NEDD1/POC5 RNAi

**Fig. 4 POC5, augmin and γTuRC promote centriole stability. a** Mitotic control and POC5 RNAi U2OS cells treated with STLC for 18 h and stained for centrin, POC5 and DNA. **b** Control and HAUS6 RNAi treated as in (**a**) and stained for centrin, HAUS6 and DNA. **c** Percentage of control or POC5 RNAi U2OS cells as in (**a**) with ≥ 4 centrioles. Data points from 3 independent experiments, 239-337 cells per condition and experiment ($p < 2.22e-16$, generalized linear model with binomial distribution). **d** Percentage of control or HAUS6 RNAi U2OS cells as in (**b**) with ≥4 centrioles. Data points from 4 independent experiments, 232–335 cells per condition and experiment ($p < 2.22e-16$, generalized linear model with binomial distribution). **e** Percentage of control or HAUS6 RNAi U2OS cells with ≥4 centrioles treated as depicted with or without BI2536. Data points from 3 (0.5 h, 2 h, 24 h + BI2536) or 6 (24 h) independent experiments, 115-350 cells per condition and experiment ($p$ (0.5 h) = 0.915; $p$ (2 h) = 1.2442e-05, $p$ (24 h) < 2.22e-16, $p$ (24 h, BI2536) = 0.0054128, generalized linear model with binomial distribution). **f** Control and GCP4 RNAi U2OS cells treated and stained as indicated. **g** Percentage of cells as in (**f**) with ≥4 centrioles after 0.5 h and 24 h in mitosis. Data points from 4 independent experiments, 100-140 cells per condition and experiment ($p$ (0.5 h) = 0.2723, $p$ (24 h) < 2.22e-16, generalized linear model with binomial distribution). **h** Control and HAUS6 RNAi U2OS cells as in (**a**) and stained as indicated. **i** Percentage of cells as in (**h**) with ≥4 centrin foci or ≥2 centrobin foci. Data points from 3 independent experiments, 300-504 cells per condition and experiment ($p$ (centrin) < 2.22e-16, $p$ (centrobin) = 0.018855, generalized linear model with binomial distribution). Horizontal lines and error bars in (**c,d,e,g,i**) depict means and standard deviations, respectively. Scale bars, 10 μm (cells) or 1 μm (magnifications). ***$p < 0.001$; *$p < 0.05$; n.s., not significant. Source data are provided as a Source Data file.

G1 cells (Figs. 2b–i, 3d, Supplementary Fig. 2a, b, d): ~52 h after siRNA transfection, 10 μM STLC was added to the culture medium for ~18 h to arrest cells in mitosis. Mitotic cells were collected, released into G1 by addition of 10 μM RO-3306 and fixed ~4 h later. Prolonged mitotic arrest assay without previous synchronization in control or HAUS6/POC5 RNAi cells (Fig. 4a–d, h, i, Supplementary Fig. 4d, e): ~52 h after siRNA transfection, 10 μM STLC was added to the culture medium for ~18 h before cells were fixed. Prolonged mitotic arrest assay in control or HAUS6 RNAi parental hTERT RPE1 cells (Supplementary Fig. 4c): cells were transfected with siRNA and 5 h later serum-starved for ~73 h. Subsequently, cells were released into a medium containing FBS and STLC for ~25 h before cells were fixed. Prolonged mitotic arrest assay with defined time points in control or HAUS6/GCP4 RNAi cells (Fig. 4e–g): ~52 h after siRNA transfection, 10 μM RO-3306 was added to the culture medium for ~18 h to arrest cells in G2. RO-3306 was washed out and 10 μM STLC was added to arrest cells in mitosis for defined time points before cells were fixed. In some cases, 100 nM BI2536 was added together with STLC. Generation of polyploid control or HAUS6 RNAi U2OS cells by inhibition of Aurora kinases (Fig. 5a–d): ~24 h after siRNA transfection, 1 μM VX-680 was added to the culture medium for ~72 h before cells were fixed. Generation of polyploid control or HAUS6 cKO hTERT RPE1 p53 KO cells by inhibition of Aurora kinases (Fig. 5f–j): Cas9 was induced with 1 μg/ml doxycycline for 72 h. At ~48 h after Cas9 induction, 1 μM VX-680 was added to the culture medium for ~72 h before cells were fixed. Induction of ciliogenesis in control or POC5 RNAi parental hTERT RPE1 or hTERT RPE1 p53 KO cells (Fig. 6a, b): cells were transfected two times with siRNA (second transfection after ~48 h) and serum-starved after a total of ~80-96 h for 48 h before cells were fixed. Induction of ciliogenesis in WT or AII-1 fibroblasts (Fig. 5c–h): cells were serum starved for 48 h before fixation. Induction of ciliogenesis in control or GCP4 RNAi hTERT RPE1 p53 KO cells (Supplementary Fig. 5a–c): cells were transfected with siRNA oligos and 4 days later serum-starved for 48 h before fixation. Induction of ciliogenesis in control or GCP4 hTERT BJ cells (Supplementary Fig. 5d, e): cells were transfected with siRNA and 24 h later serum-starved for 48 h before fixation.

**Generation of mouse strains**. A neuronal specific *Haus6* conditional KO mouse strain was generated by crossing *Haus6* floxed (*Haus6*fl) mice[83] (RBRC09630, Accession No. CDB1354K with B6.Tg(Actl6b-Cre)4092Jiwu/J mice (Jackson Laboratories). Mouse strains were maintained on a mixed C57BL/6 background in strict accordance with the European Community (2010/63/UE) guidelines in the Specific-Pathogen Free (SPF) animal facilities of the Barcelona Science Park (PCB). Mice were kept at 22–24 °C and a relative humidity of 45-65%, under IVC lighting. All protocols were approved by the Animal Care and Use Committee of the PCB/University of Barcelona (IACUC; CEEA-PCB) and by the Departament de Territori I Sostenibilitat of the Generalitat de Catalunya in accordance with applicable legislation (Real Decreto 53/2013).

**Mice genotyping**. DNA was extracted from tail biopsies by digesting biopsies with 0.4 mg/mL Proteinase K in 10 mM Tris-HCl, 20 mM NaCl, 0.2% SDS, and 0.5 mM EDTA overnight at 56 °C, followed by DNA precipitation with isopropanol. Genotyping was performed by PCR using the following primers: mAug6KO_FW and mAug6KO_Rev to detect *Haus6* WT, *Haus6* floxed, and *Haus6* KO alleles; 26994 and 30672 to detect the Cre-recombinase transgene, primers olMR7338, and olMR7339, were used as internal PCR controls. Primer sequences can be found in Supplementary Table 1.

**Neuron cell culture**. For obtaining embryonic hippocampal tissue, timed pregnant mice were sacrificed by cervical dislocation. Cell cultures were prepared from e17.5–18.5 mouse embryos (male and female) as described previously[48]. Briefly, tissue was dissected in Hank's solution, incubated in 0.25% trypsin and 1 mg/ml DNAse for 15 min at 37 °C, and dissociated into single cells by gentle pipetting. Cells were seeded on poly-D-lysine-coated glass coverslips in DMEM with 10%

FBS. Two hours after plating, the medium was replaced with a Neurobasal medium with 0.6% Glucose, 2% B27, Glutamax, and cells were kept in a 37 °C incubator with 5% $CO_2$ and a humidified atmosphere. At 3 DIV, 1 μM cytosine arabinoside was added to the medium.

**Immunofluorescence microscopy (IF) and expansion microscopy (ExM)**. Cells were grown on poly-L-lysine- or poly-D-lysine- (neurons) coated coverslips and fixed with methanol at −20 °C for a minimum of 15 min or with 3.7% paraformaldehyde at 37 °C, followed by methanol at −20 °C as described for the microtubule regrowth assay. To visualize centrioles with α-tubulin or acetylated α-tubulin, cells were incubated on ice for 30-40 min to depolymerize cytoplasmic microtubules. To remove cytoplasmic background (stainings of centrioles for α-tubulin or EGFP-HAUS6) cells were pre-extracted in ice-cold PHEM (60 mM PIPES, 25 mM HEPES, 10 mM EGTA, 2 mM $MgCl_2$) pH 6.9 with 0.1% Triton X-100 for 1–2 min before fixation. Fixed cells were washed with PBS and blocked in PBS-BT (PBS, 3% BSA, 0.1% Triton X-100) for 1 h at RT, followed by the incubation with primary antibodies in PBS-BT either for 1 h at RT or overnight at 4 °C. After washes in PBS-T (PBS, 0.1% Triton X-100) cells were incubated with secondary antibodies and 0.5 μg/ml DAPI (where appropriate) in PBS-BT for 1 h at RT. Cells were washed in PBS-T and either mounted in ProLong Gold Antifade (Thermofisher) on glass slides or further processed for ExM[36]: cells were washed with PBS and subsequently incubated in 0.1 mg/ml Acryloyl X in PBS at RT overnight. Cells were expanded in monomer solution (1 x PBS, 2 M NaCl, 2.5% acrylamide, 0.15% methylenbisacrylamide, 8.625% sodium acrylate and embedded in monomer solution containing 0.2% APS, and 0.2% TEMED. Gels were polymerized for 2 h at 37 °C and then digested with 8 U/ml Proteinase K in 50 mM Tris-HCl pH 8, 1 mM EDTA, 1 M NaCl, 0.5% Triton X-100 for 4 h at 37 °C. Gels were expanded in water (expansion factor ~4) and mounted on poly-L-lysine-coated coverslips for imaging. Based on acetylated α-tubulin staining, which yielded the most robust centriole labeling, we determined the dimensions of centrioles expanded by this method in three independent gels, measuring 44-50 centrioles per gel. Considering the respective expansion factor, we determined an average length of ~355 nm and an average width of ~190 nm. These values are very similar to values obtained for U-ExM of polyE-tubulin-stained centrioles in U2OS cells[84].

**Ultra expansion microscopy (U-ExM)**. For U-ExM[41] cells were either grown on poly-L-lysine-coated coverslips and fixed with methanol at −20 °C for a minimum of 15 min or they were grown on uncoated coverslips and not fixed prior to incubation in formaldehyde/acrylamide (Figs. 3c, 5a–d) as described below. To visualize centrioles with α-tubulin, cells were incubated on ice for 30 min to depolymerize cytoplasmic microtubules and pre-extracted in PBS with 0.1% Triton X-100 for 90 sec before fixation. Fixed cells were washed with PBS and incubated in PBS with 1.4% formaldehyde and 2% acrylamide for 5 h at 37 °C. Subsequently, coverslips were removed from the solution and embedded in gel solution (1 x PBS, 23% w/v sodium acrylate, 10% w/v acrylamide, 0.1% methylenbisacrylamide, 0.5% APS, 0.5% TEMED). Gels were polymerized for 1 h at 37 °C in a humified chamber and either stored in PBS or directly transferred into digestion buffer (200 mM SDS, 200 mM NaCl, 50 mM Tris pH 9.0). Gels were incubated for up to 15 min in digestion buffer at RT, followed by an incubation in the same buffer at 95 °C for 90 min. Gels were transferred into water to remove digestion buffer, resulting in their expansion. Gels were shrunk by incubation in PBS, followed by the incubation in blocking solution (PBS, 0.1% Tween, 0.2% BSA) for 30 min. Consequently, gels were incubated in a blocking solution containing the respective primary antibodies for either 2.5 h at RT or overnight at 4 °C, while shaking. If not indicated otherwise, antibodies were used in the same concentration as for regular IF. Gels were washed in blocking solution several times, before being incubated in blocking solution containing the respective secondary antibodies for either 2.5 h at RT or overnight at 4 °C, while shaking. Gels were washed in PBS with 0.1% Tween, expanded to their final size in water, and mounted on poly-L-lysine-coated coverslips for imaging.

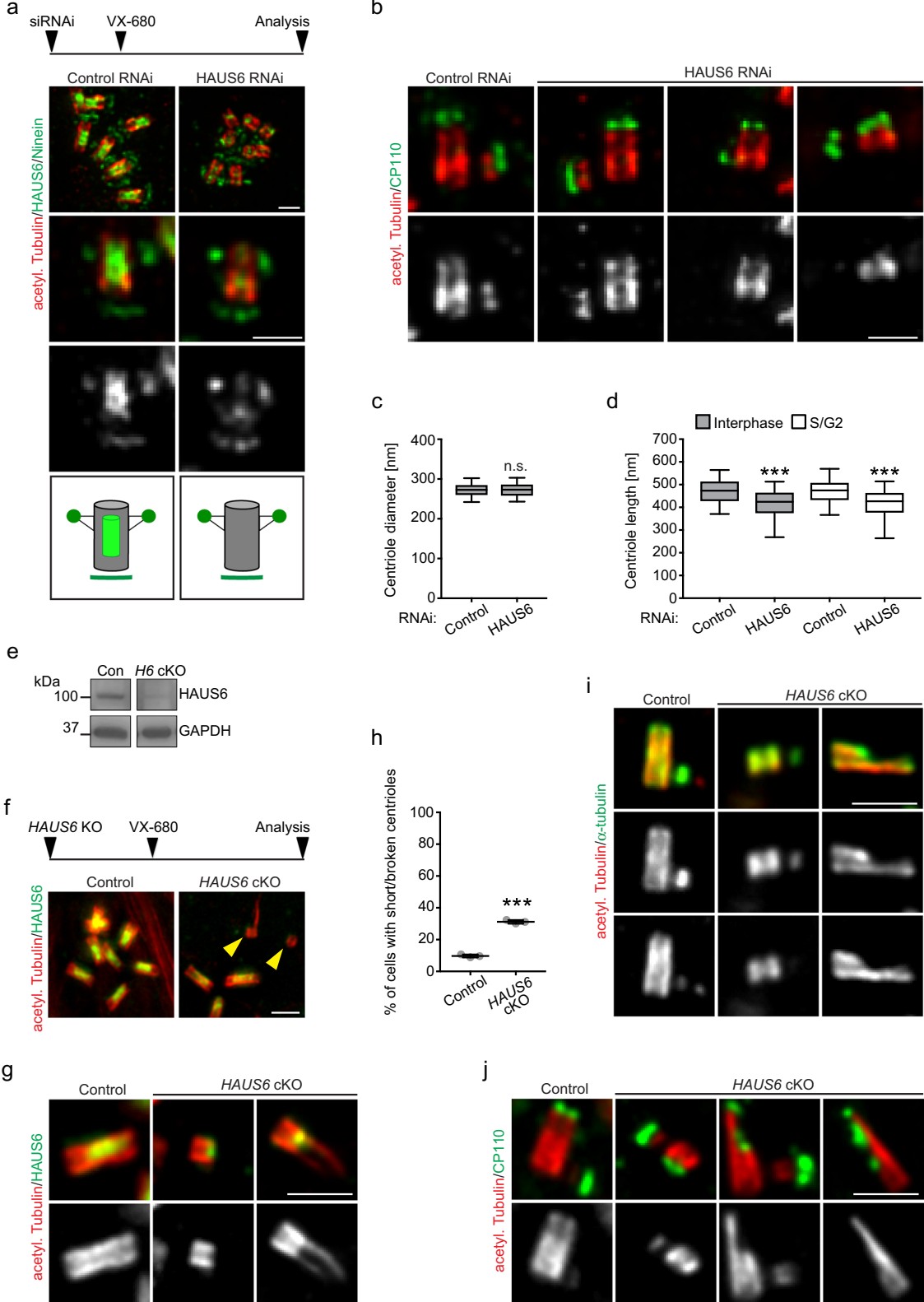

**Image acquisition and processing**. Images were acquired with an Orca AG camera (Hamamatsu) on a Leica DMI6000B microscope equipped with a 1.4 NA 100× HCX Plan Apo oil immersion objective, filter sets L5 (Leica), TX2 (Leica), DAPI ET (F46-900; AHF analysentechnik), and light source EL6000 (Leica). AF6000 software (Leica) was used for image acquisition and blind deconvolution. Alternatively, images were acquired with an MRm camera (Zeiss) on an Axiovert 200 M (Zeiss) using a 1.4 NA 63× Plan Apo oil immersion objective (Zeiss; Fig. 6c

and Supplementary Fig. 5d) or a 1.4 NA 100× UPLSApo oil immersion objective (Olympus; Fig. 6f) using Axiovision software and the following filters: Ex G 365 nm–Em BP 445/50 nm, Ex BP 475/40 nm–Em BP 530/50 nm, Ex BP 546/12 nm–Em LP 590 nm and light source: HBO 103 W/2. Images were deconvolved using the Huygens software (Scientific Volume Imaging). Images presented in Fig. 5a, b were acquired with an Orca-flash4 V2 + CMOS camera (Hamamatsu) on a Leica Dmi8 equipped with a Yokogawa CSU-X1 5000 rpm spinning disc unit and

**Fig. 5 Augmin promotes proper centriole length and architecture. a** Control RNAi and HAUS6 RNAi U2OS cells treated as depicted and stained for acetylated α-tubulin, HAUS6 and ninein. The cartoon illustrates centriole lumen localization of HAUS6 (bright green) and subdistal appendage localization of ninein (dark green). **b** Cells treated as in (**a**) and stained for acetylated α-tubulin and CP110. **c** Quantifications of the centriole diameter in cells as in (**a**). The box plot contains pooled data from 4 independent experiments, 24-64 centrioles measured per condition and experiment ($p = 0.5082$, mixed-effects linear model). **d** Centriole length in cells as in (**a**) for centrioles throughout interphase or only during S/G2 phase. The box plot contains pooled data from 4 independent experiments, 82-237 (interphase) and 13-105 (S/G2) centrioles measured per condition and experiment ($p$ (interphase) <2.22e-16; $p$ (S/G2) = 1.4954e-09, mixed-effects linear model). Boxes in (**c**) and (**d**) extend from the 25th to the 75th percentile, the horizontal line indicates the median, whiskers depict the 10th and 90th percentiles. **e** Western Blot analysis of HAUS6 in extracts from control (Con) and *HAUS6* cKO (*H6* cKO) RPE1 *p53* KO cells, 3 days after *HAUS6* KO induction. GAPDH was used as loading control. **f** Control and *HAUS6* cKO RPE1 cells treated as depicted and stained for acetylated α-tubulin and HAUS6. Yellow arrowheads point to short and broken centrioles that lack HAUS6 in the lumen. **g** Control and *HAUS6* cKO RPE1 cells treated as in (**f**) and stained for acetylated α-tubulin and HAUS6. **h** Percentage of cells as in (**f**) with short or broken centrioles. Data points are from 3 independent experiments, 259-377 cells per condition and experiment. Horizontal lines and error bars depict means and standard deviations, respectively ($p < 2.22e-16$, generalized linear model with binomial distribution). (**i**), (**j**) Control and *HAUS6* cKO RPE1 cells treated as in (**f**) and stained for acetylated α-tubulin and α-tubulin (**i**) or for acetylated α-tubulin and CP110 (**j**). Acetyl. Tubulin, acetylated α-tubulin. ***$p < 0.001$; n.s., not significant. Scale bar (all panels), 2 μm. Source data are provided as a Source Data file.

a 1.4 NA 63× HC PL Apo objective, oil (Leica) with a 0.8× lens using Metamorph software (Molecular Devices) and the following filters: 488 nm (150 mW)–Em 525/15 nm, 561 nm (100 mW), Em 607/18 nm and light source: four colors ILE laser diode (Andor technology). Images were deconvolved using the Huygens software (Scientific Volume Imaging).

Images were processed (color assignment, cropping, adjustments of overall grey levels) in ImageJ and Photoshop (Adobe) and represent maximum projections of a deconvolved stack or single sections.

For super-resolution microscopy, we imaged cells using fast random illumination microscopy (RIM) with a binary phase optical modulator (QXGA SLM Fourth Dimension) using a 100× objective lens (CFI SR APO 100XH ON 1,49 DT 0,12 Nikon) mounted on a Nikon Eclipse Ti-E and two Hamamatsu SCMOS cameras (ORCA FLASH FUSION) aligned for two-color imaging. A series of 400 3D Speckle illuminations was made on the object plane thanks to the random phase modulation from the SLM conjugate with the pupil plane of the microscope objective. A 200 nm Z interval was acquired on each image. Super-resolution images were processed by a variance matching process using random speckle illuminations with wiener parameter equal to 0.008 and 35 iterations. This method reduces drastically chromatic and spherical aberrations when compared to conventional 3D SIM[53].

**Quantification of fluorescence intensities**. Image J was used for the quantification of fluorescence intensities. Images were acquired with constant exposure settings and pixel grey levels of the focused z plane were measured within a region of interest (ROI) encompassing a single centriole or in the case of Supplementary Fig. 5c a centriole pair. Background fluorescence was measured adjacent to the ROI and subtracted.

**Length measurements**. All length measurements were performed in Image J on single z sections. Centriole length and GCP4 signal length: Only centrioles with the longitudinal axis parallel to the xy plane were analysed. The lengths were measured using a line scan drawn along the longitudinal axis in the middle of the centrioles (at mid-width and on a single z section at mid-height). The centriole length was measured based on the acetylated α-tubulin staining. For GCP4, when the signal appeared dotted (see Fig. 6d, cyan arrowheads), the length was quantified as the sum of the lengths of the individual foci. Measured values were divided by the gel expansion factor to obtain real values (Figs. 5d, 6g). GCP4 lumen coverage corresponds to the GCP4 length expressed as a % of the centriole length, for each individual centriole. Centriole diameter: the diameters were measured on centrioles (acetylated α-tubulin) with the longitudinal axis perpendicular to the xy plane. Measured values were divided by the gel expansion factor to obtain real values. Centrin-centrin distance: the distance between centrin foci of engaged mother and daughter centrioles was determined by measuring the length between the most proximal signals using a line scan as depicted in Fig. 1g. Measured values were normalized by setting the condition 'HAUS6 absent' or 'NEDD1 absent' in each experiment to 1. Each experiment corresponds to one gel and obtained values are thus independent of the gel expansion factor.

**Western blotting (WB)**. Cells were washed in PBS and lysed in 50 mM HEPES, pH 7.5, 150 mM NaCl, 1 mM $MgCl_2$, 1 mM EGTA, 0.5% NP-40, and protease inhibitors for at least 10 min on ice. The extract was cleared by centrifugation and subjected to SDS PAGE, followed by the transfer of proteins to PVDF membranes by tank blotting. Subsequently, membranes were blocked and probed with antibodies.

**Antibodies**. Generation of the rabbit polyclonal antibodies against HAUS6 (WB, 1:2000; IF, 1:1000 or 1:500 (ExM and U-ExM)) has been described previously[85]. To generate a polyclonal anti-GCP4 antibody (IF, 1:200 or 1:100 (ExM)), a His-tagged N-terminal fragment of the human GCP4 (AS 1–347) was expressed in ArticExpress cells (Agilent Technologies), solubilized in 8 M urea and affinity-purified using Ni-Sepharose beads. The protein was used for immunization of rabbits (Antibody Production Service, Facultat de Farmàcia, Universitat de Barcelona, Spain). Anti-GCP4 antibody was affinity-purified using the antigen subjected to PAGE and blotted on membranes. Other antibodies used in this study were: mouse anti-γ-tubulin (TU-30, Exbio; IF, 1:500 or 1:250 (ExM)), rabbit anti-γ-tubulin R75[86] (1:1000 or 1:250 (U-ExM)), rabbit anti-α-tubulin (ab18251, Abcam; 1:300 or 1:500 (ExM and U-ExM)), mouse anti-acetylated α-tubulin (clone 6-11B-1, Merck; IF, 1:1000, 1:500 (U-ExM) or 1:250 (ExM)), mouse anti-polyglutamylated tubulin (GT335, AdipoGen; ExM, 1:250), rabbit anti-NEDD1[22] (ExM, 1:250), rabbit anti-HAUS5[37] (ExM, 1:100), rabbit anti-pericentrin[22] (ExM, 1:250), rabbit anti-GFP (A6455, Invitrogen; ExM, 1:250), chicken anti-GFP (GFP-1020, Aves Labs; ExM, 1:250), mouse anti-centrin 1 (clone 20H5, Millipore; IF, 1:500 or 1:250 (ExM)), rabbit anti-POC5 (A303-341A, Bethyl Laboratories; WB, 1: 2500, IF: 1:500 or 1:250 (ExM)), mouse anti-SAS-6 (sc-81431, Santa Cruz; IF, 1:100 or 1:50 (ExM)), mouse anti-centrobin[87] (IF, 1:500), rabbit anti-ninein[88] (IF, 1:100 or 1:250 (U-ExM)), rabbit anti-ODF2 (43840, Abcam; IF, 1:500), mouse anti-ARL13B (sc-515784, Santa Cruz; IF, 1:100), rabbit anti-CEP192[42] (ExM, 1:500), mouse anti-CP110 (Proteintech 12780-1-AP, U-ExM 1:250), rabbit anti-CP110 (gift by Andrew Holland, unpublished; Fig. 5j, U-ExM 1:500), rabbit anti-CEP164 antibody (Proteintech 22227-1-AP, U-ExM 1:250), mouse anti-GAPDH (sc-47724, Santa Cruz Biotechnology; WB, 1:10000). Alexa-Fluor-350-, Alexa-Fluor-488-, Alexa-Fluor-568- and Alexa-Fluor-647-conjugated, cross-adsorbed secondary goat anti-rabbit, goat anti-mouse or goat anti-chicken antibodies (Thermo Fisher, A11029, A11034, A21131, A21121, A11036, A11031, A21144, A21068, A21244, A11039) were used at 1:500 or 1:100 (ExM). Streptavidin Alexa-Fluor-594 (Invitrogen, S-11227) at 1:5000 and Horseradish-peroxidase-coupled secondary goat anti-mouse or goat anti-rabbit antibodies for WB (Jackson ImmunoResearch Laboratories, AB_10015289, AB_2313567) at 1:5000.

**Plasmids**. The EGFP-HAUS8 expression plasmid was provided by Laurence Pelletier[37]. The POC5-GFP expression plasmid was obtained from Ciaran Morrison[49]. The plasmid expressing EGFP-HAUS6 was generated by cloning HAUS6 cDNA into pCS2-EGFP using AscI and FseI restriction sites. Site-directed mutagenesis was used to render EGFP-HAUS6 RNAi-insensitive (HAUS6 591 A > G; 594 T > C; 597 G > A; 600 G > C). A plasmid for expression of BirA-HAUS6 was generated by subcloning RNAi-insensitive HAUS6 into pCDNA5 FLAG-BirA^R118G (provided by Brian Raught[89]). Subsequently, FLAG-BirA^R118G-HAUS6 was amplified by PCR and inserted into pEGFP-N1, replacing EGFP. An all-in-one plasmid for the constitutive expression of sgRNA and the doxycycline-inducible expression of Cas9, TLCV2, was a gift from Adam Karpf (Addgene plasmid #87360; http://n2t.net/addgene: 87360; RRID:Addgene _87360)[90]. To generate inducible *HAUS6* KO cells, the following guide was cloned into this vector via BsmBI sites: 5′-GGCCCGGCAACCATTGCCTG-3′.

**BioID**. U2OS cells stably expressing BirA-HAUS6 or BirA alone (negative control) were incubated in a culture medium containing 50 μM biotin overnight. Cells were then washed with PBS and scraped from the plate. After an additional washing in cold PBS, cells were lysed in 10 ml of RIPA buffer (1% Triton X-100, 50 mM Tris-HCl pH 7.5, 150 mM NaCl, 1 mM EDTA, 1 mM EGTA, 0.1% SDS, 0.5% sodium deoxycholate) containing 250 U turbonuclease (Sigma Aldrich) and protease inhibitors and incubated for 1 h at 4 °C. Pellets were sonicated and centrifuged at 16,100xg at 4 °C for 30 min. The supernatant was added to streptavidin agarose resin and incubated for 3 h at 4 °C. Beads were washed several times with 50 mM ammonium bicarbonate and finally resuspended in 100 μl of 50 mM ammonium bicarbonate.

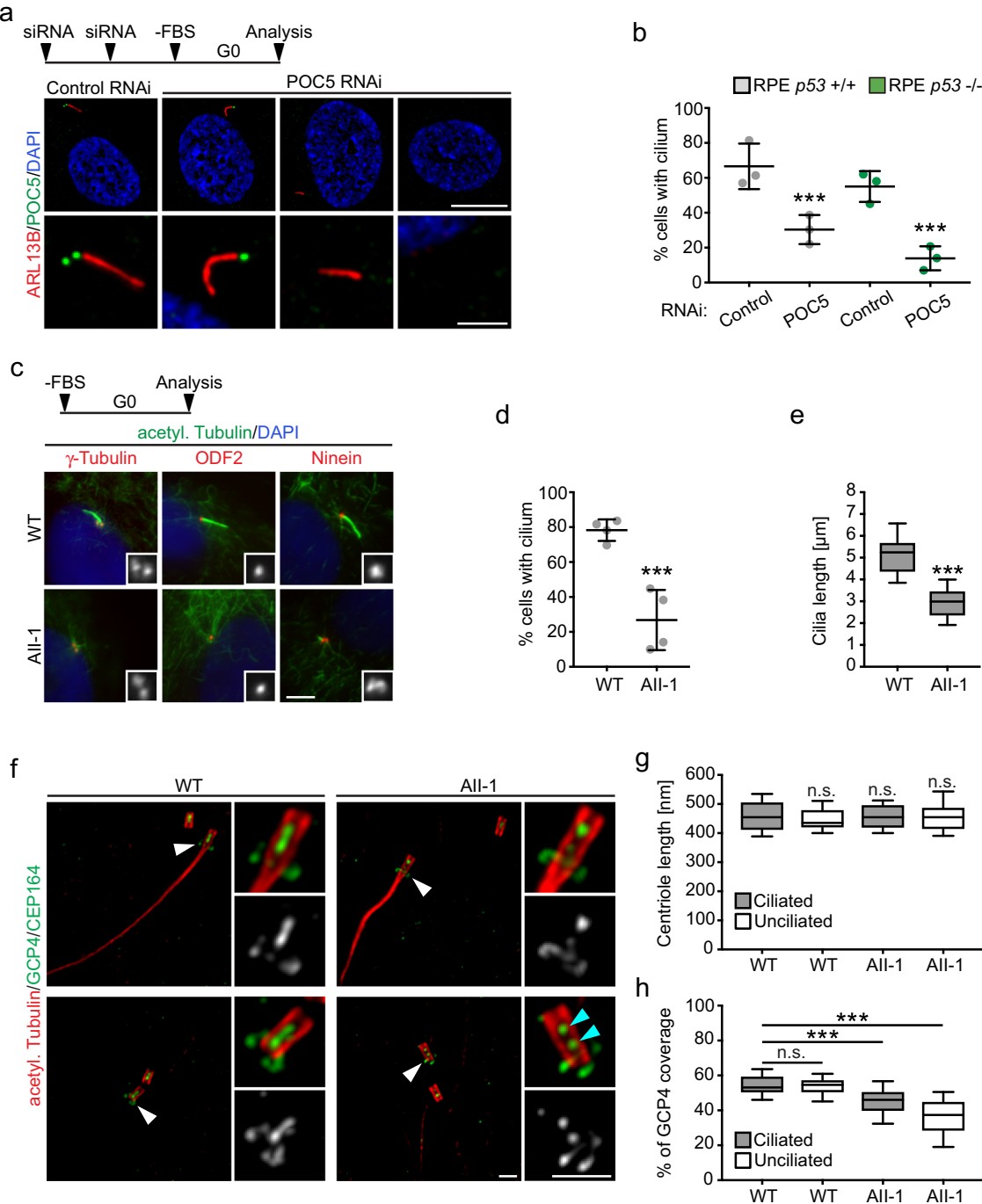

**Mass spectrometry analysis**. Samples were digested with 2 μg trypsin in 50 mM $NH_4HCO_3$ at 37 °C overnight. Additional 1 μg trypsin was added and samples were incubated for 2 h at 37 °C before formic acid was added (1% final concentration). Samples were cleaned through C18 tips (polyLC C18) and peptides were eluted with 80% acetonitrile/1% formic acid and diluted to 20% acetonitrile/0.25% formic acid before loading into strong cation exchange columns (polyLC SCX). Peptides were eluted in 5% $NH_4OH$/30% methanol. Samples were evaporated to dryness and reconstituted in $H_2O$ with 3% acetonitrile/1% formic acid in a total volume of 50 μl. For mass spectrometry analysis, the reconstituted sample was further diluted 1:8 in $H_2O$ with 3% acetonitrile/1% formic acid. Samples were injected in duplicate (5 μl per injection).

The sample was loaded at a flow rate of 15 μl/min on a 300 μm x 5 mm PepMap100, 5 μm, 100 A, C18 μ-precolumn using a Thermo Scientific Dionex Ultimate 3000 chromatographic system (Thermo Scientific). Peptide separation was done with a 90 min run on a C18 analytical column (Acclaim PepMapR RSLC 75 μm × 50 cm, nanoViper, C18, 2 μm, 100 A, Thermo Scientific), comprising three consecutive steps with linear gradients from 3 to 35% B in 60 min, from 35 to 50% B in 5 min, and from 5% to 85% B in 2 min. Isocratic elution was done at 85% B in

5 min and stabilization to initial conditions (A = 0.1% formic acid in $H_2O$, B = 0.1% formic acid in $CH_3CN$). The outlet of the column was directly connected to a TriVersa NanoMate (Advion) fitted on an Orbitrap Fusion Lumos™ Tribrid (Thermo Scientific). The mass spectrometer was operated in a data-dependent acquisition mode, survey MS scans were acquired with a resolution of 120,000 (defined at 200 m/z), and lock mass was defined at 445.12 m/z in each scan. In each scan, the top speed (most intense) ions were fragmented by CID and detected in the linear ion trap. The ion count target values for the survey and MS/MS scans were 400,000 and 10,000, respectively. Target ions already selected for MS/MS were dynamically excluded for 15 s. Spray voltage in the NanoMate source was set to 1.60 kV. RF Lens was tuned to 30%. The minimal signal required to trigger MS to MS/MS switch was set to 5000. The spectrometer was working in positive polarity mode and singly charge state precursors were rejected for fragmentation. Database searching was done with Proteome Discoverer software v2.5.0.400 (Thermo) and MaxQuant v1.6.14.0 using Sequest HT and Andromeda search engines, respectively, and SwissProt Human release 2021 01 with contaminants database and manually introduced user proteins. Searches against targeted and decoy database were used for determining the false discovery rate (FDR). Search

**Fig. 6 POC5 and γTuRC are required for ciliogenesis. a** Control and POC5 RNAi RPE1 cells treated and stained as indicated. Scale bar, 10 μm (cells) or 2 μm (magnifications). **b** Percentage of ciliated RPE1 WT and RPE1 p53 KO cells treated as in (**a**). Data points are from 3 independent experiments, 118-202 WT and 82-113 p53 KO cells per condition and experiment (p (p53+/+) < 2.22e-16, p (p53 -/-) < 2.22e-16, generalized linear model with binomial distribution). **c** Serum-starved control (WT) or patient (AII-1) fibroblasts stained for acetylated α-tubulin and either γ-tubulin, ODF2 or ninein and DNA. Insets show magnified centrosome regions. Scale bar, 5 μm. **d** Percentage of ciliated control (WT) or patient (AII-1) fibroblasts as in (**c**). Data points are from 4 independent experiments, 99-122 cells per condition and experiment (p (WT) < 2.22e-16, generalized linear model with binomial distribution). **e** Cilia length in cells as in (**c**). The box plot contains pooled data from 3 independent experiments, 13–26 cilia per condition and experiment (p < 2.22e-16, generalized linear model). **f** Serum-starved control (WT) or patient (AII-1) fibroblasts stained as indicated in U-ExM. White arrowheads indicate CEP164 distal appendage staining, cyan arrowheads mark discontinuous luminal GCP4 signal. Scale bar, 2 μm. **g** Centriole length in cells as in (**f**). The box plot contains pooled data from 3 independent experiments, 20–42 centrioles per condition and experiment (p (WT ciliated/WT unciliated) = 0.64112, p (WT ciliated/AII-1 ciliated) = 0.99348, p (WT ciliated/AII-1 unciliated) = 0.99376, mixed-effects linear model). **h** Percentage of GCP4 centriole length coverage in cells as in (**f**). The box plot contains pooled data from 3 independent experiments, 20–42 centrioles per condition and experiment (p (WT ciliated/WT unciliated) = 0.94869, p (WT ciliated/AII-1 ciliated) = 1.2652e-10, p (WT ciliated/AII-1 unciliated) <2.22e-16, mixed-effects linear model). (**b,c**): horizontal lines and error bars depict means and standard deviations, respectively. (**e,g,h**): Boxes extend from the 25th to the 75th percentile, horizontal lines indicate medians, whiskers depict the 10th and 90th percentiles. ***p < 0.001; n.s., not significant. Source data are provided as a Source Data file.

parameters for trypsin enzyme specificity allowed for two missed cleavage sites, oxidation in M and acetylation in protein N-terminus. Peptide mass tolerance was 10 ppm and the MS/MS tolerance was 0.6 Da. Peptides with q value lower than 0.01 and FDR < 1% were considered as positive with a high confidence level.

For the quantitative analysis contaminant identifications were removed and unique peptide spectrum matches of protein groups identified with Sequest HT and Andromeda were analyzed with SAINTexpress-spc v3.11[91]. High confidence interactors were defined as those with Bayesian false discovery rate BFDR ≤ 0.02 and fold change FC ≥ 3.

**Statistics and reproducibility.** Cell counts: counts of cells belonging to the positive and negative condition were used to fit a generalized linear model with a binomial distribution. The R packages 'lme4' and 'multcomp' were used to fit the model and compute raw and adjusted p-values with the 'Shaffer' method[92–94]. Length and intensity measurements: for experiments regarding cilia/centriole length, % of GCP4 centriole coverage or fluorescence intensities, a boxcox transformation was applied to ensure the normality of data. The parameter of the boxcox transformation was computed by fitting a linear model and using the function 'boxcox' from the 'MASS' R package[95]. To avoid problems with the transformation, zero values were increased by half the minimum nonzero value in the dataset. Transformed values were used to fit a random-effects linear model with experiment as a fixed covariate and technical replicates as random effects. Whenever possible the 'lmer' function from the lme4 package was used to fit the model. For experiments showing convergence problems, a Bayesian version for mixed-effects linear models was used, as implemented in the 'blmer' function from the 'blme' R package[96]. Raw and adjusted p-values were computed with the 'glht' function and method 'single-step'. The results, together with the number of independent experiments and sample sizes, are reported in the figures, figure legends, and Source Data file. For experiments other than quantitative experiments, all attempts at replication were successful. Protein localizations have been confirmed in at least two independent experiments (Figs. 1a–e, 2b–e, 3b, c, e 5i, Supplementary Figs. 1a, b, 2e, 3a, c) except for CP110 localization in U2OS (Fig. 5b) and RPE1 p53 KO (Fig. 5j) cells and POC5 and HAUS6 localization in untreated mitotic cells (Supplementary Fig. 4a, b). Dependencies have been confirmed in at least two independent experiments (Figs. 2c–g, 3d, Supplementary Fig. 2a, b, d, f) except for CEP192 localization after CEP192 RNAi (Fig. 2b). Microtubule regrowth along the centriole wall (Fig. 1b) has been confirmed in at least two independent experiments. The displayed images are representative examples. The BioID was performed once. Depletion of HAUS6 or POC5 after RNAi (Supplementary Figs. 2c, 3b) or HAUS6 depletion after HAUS6 KO induction (Fig. 5e) has been confirmed at least two times by Western Blot analysis.

**Reporting Summary.** Further information on research design is available in the Nature Research Reporting Summary linked to this article.

## Data availability
All source data associated with the results presented in this study are provided with the published article. The mass spectrometry proteomics data have been deposited to the ProteomeXchange Consortium via the PRIDE[97] partner repository with the dataset identifier PXD027352. Proteomics data are additionally provided in Supplementary Data 1. Source data are provided with this paper.

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

## Acknowledgements

NS was supported by an EMBO long-term fellowship (ALTF 820-2015) and a Marie Skłodowska-Curie Action fellowship: This project has received funding from the European Union's Horizon 2020 research and innovation programme under the Marie Sklodowska-Curie grant agreement No 703907. ID was funded by the European Union's Horizon 2020 research and innovation programme under the Marie Skłodowska-Curie grant agreement No. 754510. LH and AM were in part supported by grant 13-BSV8-0007-01 from "Agence Nationale de la Recherche" (France), and by grant SFI20121205511 from "Fondation ARC pour la recherche sur le cancer". JL acknowledges support by grants BFU2015-69275-P (MINECO/FEDER), PGC2018-099562-B-I00 (MICINN), network grants 2017 SGR 1089 (AGAUR) and RED2018-102723-T (MICIU), and by intramural funds of IRB Barcelona, recipient of a Severo Ochoa Centre of Excellence Award from the Spanish Ministry of Science and Innovation and supported by CERCA (Generalitat de Catalunya). We thank the IRB Mass Spectrometry & Proteomics Core Facility, a member of ProteoRed, PRB3-ISCIII, supported by grant PRB3 (IPT17/0019 -ISCIIISGEFI/ERDF), the IRB Biostatistics/Bioinformatics Core Facillity and the Animal Facility of the Parc Scientific de Barcelona for excellent support. We thank Hélène Dollfus (University of Strasbourg, Strasbourg, France), Meng-Fu Bryan Tsou (Memorial Sloan Kettering Cancer Center, New York, USA), Laurence Pelletier (Lunenfeld-Tanenbaum Research Institute, Toronto, Canada), Ciaran Morrison (National University of Ireland, Galway, Ireland), Brian Raught (Ontario Cancer Institute and Department of Medical Biophysics, University of Toronto, Toronto, Canada) and Andrew Holland (Johns Hopkins University School of Medicine, Baltimore, USA) for cells and reagents. We thank Julian Frädrich for initial localization studies in control and *Haus6* cKO neurons. We thank Thomas Mangeat and Vanessa Tillement from the Toulouse RIO Imaging platform (LITC) for help with microscopy.

## Author contributions

N.S. designed experimental strategies, performed most of the experiments, prepared figures, and contributed to manuscript writing. L.H. analyzed the relative localization of scaffold components by U-ExM/RIM, performed the prolonged mitotic arrest assay after GCP4 depletion, designed and performed the VX-680 experiments in U2OS cells and the ciliogenesis experiments in fibroblasts. ID performed half of the distance measurements presented in Fig. 1g, performed CEP192 RNAi to analyze the localization of γ-tubulin, POC5 and HAUS6/to measure fluorescence intensities at centrioles, and some of the POC5 RNAi experiments in RPE1 *p53* KO cells. R.V. generated conditional *Haus6* KO mice and prepared neuronal cultures. C.L. supervised animal experiments, performed the BioID experiment, and helped to establish the conditional *HAUS6* KO line. A.M. participated in experiments with patient fibroblasts. J.L. supervised the study, proposed experimental strategies, and contributed to manuscript writing.

## Competing interests

The authors declare no competing interests.
