## [Peer Review File · Nature Communications]

Sub-centrosomal mapping identifies augmin- γ TuRC as part of a centriole-stabilizing scaffoldREVIEWER COMMENTS

Reviewer #1 (Remarks to the Author):

In this manuscript Schweizer and colleagues investigate the role of Augmin-gTuRC in the stabilization of the centriole structure and cilia formation. Briefly, Using ExM the authors show that Augmin and gTuRC components form separable centrosomal population, one of which intriguingly restricted to the centriolar lumen. The authors go on to show that the recruitment of Augmin and gTuRCs occurs during centriole elongation and that luminal recruitment of NEDD1 does not require CEP192 but the recruitment of NEDD1 to the outer centriole wall does. Next, the authors show that the recruitment of gTuRCs to the centriolar lumen requires Augmin and NEDD1 in HAUS6 RNAi treated U2OS cells and in mouse hippocampal neurons where HAUS6 was knocked out post-mitotically. Using HAUS6 BioID, the authors identify POC5 as a proximal interactor and show that POC5 localizes as well to the lumen of elongated centrioles. The authors further show that luminal localization of Augmin and gTuRCs require POC5. In support of the idea that POC5 associates with Augmin and gTuRCs, the overexpression of POC5 induced filaments containing all three components. Finally, the authors show that both POC5 and Augmin promote centriole integrity and that gTuRCs and POC5 are required for ciliogenesis. Overall this is a very nice paper that will appeal to both the centrosome and cilia fields. Furthermore, this paper describes a new function for HAUS6 in the luminal recruitment of gTuRC which is both novel and exciting. The authors should consider the following issues:

Major issue:

- Do the authors know if Augmin is required for ciliogenesis? The authors mention that they could not deplete it from the mother centriole and thus could not determine this. I find this somewhat unsatisfying. What have they tried exactly? It is not mentioned but do the HAUS6 KO hippocampal neurons make cilia, I am not an expert but my quick perusal of the literature suggest that these cells have cilia, so what happens in the HAUS6 KO? It seems to me that the authors need to close the loop and show that Augmin is indeed required for ciliogenesis if they want to claim that luminal recruitment of gTuRC is.
- All the experiments implicating Augmin in luminal recruitment were done with HAUS6 RNAi and KO cells. That data is rather convincing. However, the authors argue that Augmin is required. To substantiate this statement, the authors need to show that other Augmin subunits are implicated in this process as it remains possible, with the data shown, that only HAUS6 is implicated. Consistent with this notion, HAUS6 is the NEDD1 binding component of Augmin, which in turn recruits gTuRCs and it thus remains possible, that the luminal function of Augmin is carried out by HAUS6 alone.
- Do the authors have any evidence indicating that centriolar POC5, gTuRC and Augmin subunits (not only HAUS6) form a complex in vivo at endogenous levels? The BioID data and overexpression (filaments) data alone can entirely support that claim while biochemical evidence could.

Minor issues:

- CEP192 depletion does not effect NEDD1 luminal localization. The authors should confirm that POC5, HAUS6 and gTuRC luminal localization is also unaffected as this would be consistent with their model but is not shown.
- Raw BioID mass spec data should be provided in full.

Reviewer #2 (Remarks to the Author):

In this paper, Schweizer et al. identify a non-canonical role for Augmin- γ -TuRC in maintaining centriole integrity to allow efficient ciliogenesis. They show that during centriole duplication, γ -TuRCs are recruited first to the outer centriole wall via CEP192 and later to the centriolar lumen via Augmin and Poc5. Both sites of γ -TuRC localisation require the γ -TuRC tethering factor NEDD1, which has been previously shown to mediate Augmin-dependent recruitment of γ -TuRCs to the sides of pre-existing microtubules and to mediate γ -TuRC recruitment via CEP192 to the PCM during mitosis. The authors used BioID to identify Poc5 as an Augmin interactor and then showed that depletion of Poc5 leads to a loss of Augmin- γ -TuRC within the centriole lumen. To assess a role for luminal Augmin- γ -TuRC the authors depleted Poc5 and showed that fewer centrioles remain after an extended mitotic arrest, suggesting that centriole integrity is impaired after loss of luminal Poc5 and Augmin- γ -TuRC. The authors also showed that the ability to form cilia is impaired after Poc5 depletion, suggesting that centriole integrity defects caused by the loss of luminal Augmin- γ -TuRC perturb cilia formation. They also observed cilia defects in a patient with a mutation in GCP4 (a γ -TuRC protein), suggesting that the cilia defects in this patient may be attributed to decreased centriole stability caused by a reduction in γ -TuRCs within the centriolar lumen.

This is an interesting paper that expands the range of γ -TuRC function and provides insights into the γ -TuRC-related cause of human disease. However, I feel that prior to publication some links between observations and conclusions need to be supported by further data and I outline this below.

Major comments

1) At present, the authors conclude that centriole integrity is affected after the loss of Augmin- γ -TuRC from the centriole lumen and that this leads to the observed reduction in cilia / cilia defects, yet the authors lack direct evidence for this. Can the authors use expansion microscopy to identify centriole integrity defects (prior to loss of centrin that occurs only after a prolonged mitotic arrest) after Poc5 or HAUS6 depletion, perhaps by using acetylated tubulin and centrin antibody staining? Centriole width, or the distance between the centrin and the centriole wall, could increase, for example. Can the authors assess the structure of centrioles/basal within Poc5 depleted cells that have failed to generate cilia or that have shorter cilia?

2) The authors link the patient mutation in GCP4 to cilia defects, hypothesising that these cilia defects are caused by centriole integrity defects. Can the authors recapitulate the cilia defects observed in GCP4 mutant patient fibroblasts by depleting GCP4 from RPE1 cells? The model would suggest that centriole integrity is also be perturbed in the extended mitosis assay after GCP4 depletion – can the authors show this?

3) This relates somewhat to point 2. Is it Augmin or γ -TuRC, or both, that is necessary for centriole integrity? The authors suggest that the Y and L shaped structures (from EM images) previously observed to link centriole wall microtubules may represent Augmin, while ring structures in the lumen may represent γ -TuRCs. This is hard to reconcile with a model where Augmin binds and recruits γ -TuRCs to the lumen, as in this model one would expect Augmin and γ -TuRCs to be in very close proximity to each other. The EM observations may also suggest that Augmin is more important than γ -TuRC for centriole integrity. Depleting γ -TuRC components and assessing centriole integrity using the extended mitosis assay could help answer this, especially as this provides a reliable quantifiable readout of centriole integrity. It would then be interesting to show whether Augmin is still present after GCP4 or NEDD1 (i.e. γ -TuRC) depletion.

Minor comments

4) Microtubule nucleation assay in Figure 1 – please point out that this is during interphase (I assume it is).

5) Figure 2: the images of localisation defects after RNAi knockdowns would benefit from some form of quantification – either fluorescence intensity measurements, or fraction of cells without staining.

Reviewer #3 (Remarks to the Author):

Sub-centrosomal mapping identifies augmin-gTuRC as part of a centriole-stabilizing scaffold by Schweizer et al. is an interesting paper presenting results mostly based around expansion microscopy of labelled centrioles under a number of conditions. They use the resultant data to draw significant new conclusions on centriole biology an vital cellular component involved in many biological processes.

Overall I think the paper is interesting and has useful new results for the field however I have significant concerns that several of the statistical results presented in the paper are overstated.

Figure 1g presents values of:

HAUS6 (absent): $0.5 \pm 0.2 \mu\text{m}$, HAUS6 (lumen): $0.8 \pm 0.2 \mu\text{m}$

and states:

p (HAUS6 absent/lumen) < 0.01

Yet if the absent value is 1 SD lower (chance $\sim 15\%$) and the lumen value is 1 SD higher (chance $\sim 15\%$) then the difference will be reversed, this has a chance of roughly $.15 \cdot .15$ or > 0.02 .

Several other comparisons have suspiciously high significance given the means and SD's quoted. These include Fig 4e, 24h data, Fig 5 e and Fig S2c. This last one is particularly strange, a percentage of intensity, which cannot be below 0, has a value of $100 \pm 50\%$ and yet a significance of < 0.001 . I don't believe that this data can be normally distributed and that significance is just mad.

Other more minor issues:

The paper relies extensively on the data from Expansion microscopy and yet the expansion factor ($\sim 4x$) is only cited in the methods section, and evidence for its validity is completely absent. The centriole is a well studied structure and simple measurements from their existing data measuring, for instance, the diameter of the microtubule structures could be compared to existing literature in optical microscopy and EM to give a real value.

I feel the introduction could do with more of a summary of existing state of the field regarding gamma-TuRC and its components and interactors as this information is not gathered together anywhere but scattered throughout the various results sections.

Fig 1g is reliant on very few data points, one experiment and 9-13 measurements per condition. This experiment needs to be repeated to generate more data to sustain the claimed significance, also see the

point about the statistics above.

The description of the imaging is incomplete, it needs information about the light sources and filters used.

The image processing section says "Images were processed in ImageJ and Photoshop (Adobe) and represent maximum projections of a deconvolved stack or a single section, respectively" I don't understand what the respectively refers to and how was photoshop used to process the images?

Several pieces in the methods refer to centrifugation speeds in rpm, these should be given as g values in order for people to be able to reproduce the methods.

In the Mass spec methods section the authors say

"Samples were cleaned through C18 tips (polyLC C18) and peptides were eluted with 80% acetonitrile/1% formic acid and consequently diluted to 20% acetonitrile/0.1% formic acid before loading into strong cation exchange columns (SCX)"

How do they dilute from 80:1 to 20:0.1, the ratios of the concentrations are totally different.

"Samples were dried and reconstituted in H₂O with 3% acetonitrile/1% formic acid (1:8) in a total volume of 50 µl." how does 3:1% relate to the 1:8 in brackets?

I was particularly impressed with the cartoons associated with every centriole image which made it much easier to understand the orientation and context of each image.

Reviewer #4 (Remarks to the Author):

This manuscript by Schweizer et al. describes the use of expansion microscopy to map the different populations of centrosomal γ TuRC, identify the augmin-mediated localization to the centriole lumen by interaction with POC5, and demonstrates that disruption of this luminal localization leads to centriolar and ciliary defects. I have focused my comments on the application of BioID to HAUS6 to identify interacting proteins in the centriole.

Specific comments:

This is a slightly unconventional approach to BioID that first biochemically separates the centrosomal material from the rest of the cell prior to biotin-affinity pulldowns. There is nothing wrong with this approach other than likely excluding associations that occur at sites distal to the centrosomes and/or the loss of proteins that are less strongly associated with the centrosome. Perhaps some language could be included about how this approach would exclude such associations. Western blots showing biotinylation of whole cells vs isolated centrioles could also help clarify what is being analyzed.

There appears to be a lack of a control that is typically used for BioID experiments. Most often this is the expression of the ligase that is not fused to anything, or is fused to a small targeting motif to force

it to a specific subcellular compartment (e.g. a NLS for the nucleus). The purpose of this control is to exclude the many false positive associations that can occur due to affinity between various cellular proteins and the ligase. If the BioID-fused to the protein of interest detects a protein uniquely or at least substantially more than it is detected by the ligase-alone, then the association is considered valid. In this case a cell line not expressing any BioID was used as a control, which is a good control for non-specific associations with the streptavidin-coupled matrix or naturally biotinylated proteins but not for proteins with variable affinity to the BioID ligase of which there are many. Since there is essentially no reporting of considerable numbers of other proteins associated with the bait and the functional association of HUS6 and POC5 are validated with other approaches it may not be necessary to repeat these BioID experiments with the proper controls; however, the limitations of the BioID experiment as performed should be noted.

Reviewer #1 (Remarks to the Author):

In this manuscript Schweizer and colleagues investigate the role of Augmin-gTuRC in the stabilization of the centriole structure and cilia formation. Briefly, Using ExM the authors show that Augmin and gTuRC components form separable centrosomal population, one of which intriguingly restricted to the centriolar lumen. The authors go on to show that the recruitment of Augmin and gTuRCs occurs during centriole elongation and that luminal recruitment of NEDD1 does not require CEP192 but the recruitment of NEDD1 to the outer centriole wall does. Next, the authors show that the recruitment of gTuRCs to the centriolar lumen requires Augmin and NEDD1 in HAUS6 RNAi treated U2OS cells and in mouse hippocampal neurons where HAUS6 was knocked out post-mitotically. Using HAUS6 BioID, the authors identify POC5 as a proximal interactor and show that POC5 localizes as well to the lumen of elongated centrioles. The authors further show that luminal localization of Augmin and gTuRCs require POC5.

In support of the idea that POC5 associates with Augmin and gTuRCs, the overexpression of POC5 induced filaments containing all three components. Finally, the authors show that both POC5 and Augmin promote centriole integrity and and that gTuRCs and POC5 are required for ciliogenesis. Overall this is a very nice paper that will appeal to both the centrosome and cilia fields. Furthermore, this paper describes a new function for HAUS6 in the luminal recruitment of gTuRC which is both novel and exciting. The authors should consider the following issues:

Major issue:

- Do the authors know if Augmin is required for ciliogenesis? The authors mention that they could not deplete it from the mother centriole and thus could not determine this. I find this somewhat unsatisfying. What have they tried exactly?

A major technical issue is that augmin siRNA treatment causes mitotic delay/arrest due to impaired spindle assembly before luminal augmin can be depleted. To overcome this issue and avoid mitotic delay/arrest, we have tried RNAi-mediated depletion of different augmin subunits in combination with various synchronization protocols. However, in conditions where cells cycle less, augmin depletion was always inefficient, especially in the centriole lumen. We also tested inducible HAUS8 knockout cells that we obtained from the Cheeseman lab (McKinley and Cheeseman, 2017), but this did not solve the above issues. We eventually discovered that overriding the spindle assembly checkpoint by treatment with the aurora kinase inhibitor VX-680 allowed augmin depleted U2OS cells to progress through mitosis and continue cycling. In the presence of inhibitor cells progressed through mitosis without division and accumulated extra centrioles, and some of these centrioles were depleted of augmin (see below). Unfortunately, in RPE1 cells this protocol seemed to be too toxic and did not work well to obtain mature mother centrioles lacking HAUS6, neither in the wildtype nor a p53 KO background. For this reason, we could not test ciliogenesis.

It is not mentioned but do the HAUS6 KO hippocampal neurons make cilia, I am not an expert but my quick perusal of the literature suggest that these cells have cilia, so what happens in the HAUS6 KO?

This is a good point and in fact we tried this. Unfortunately, conditional knockout of augmin (Haus6) in mouse apical progenitors is embryonic lethal and causes loss of cortical and hippocampal structures at early embryonic stages. We show this in another manuscript posted on bioRxiv (Viais et al., 2020). This prevented us from obtaining neurons from these animals for culturing and studying ciliogenesis. When the knockout is induced in early differentiating neurons, as in the current manuscript, luminal augmin is still detected at 2 days of in vitro culture, but is eventually depleted after 9 days (to illustrate this we have now included also the earlier time point; Suppl. Fig. 2e,f,g). However, ciliogenesis occurs much earlier, starting at day 1, before HAUS6 is depleted. Most neurons are ciliated a few days later (Berbari et al., 2007; Miki et al., 2019). Thus, in this system we did not have means to remove luminal augmin before ciliogenesis starts.

It seems to me that the authors need to close the loop and show that Augmin is indeed required for ciliogenesis if they want to claim that luminal recruitment of gTuRC is.

As explained above, the importance of augmin for mitosis and its delayed depletion in cultured, post-mitotic neurons, prevented us from directly testing augmin's role in ciliogenesis. However, we show that depletion of POC5, which recruits augmin and γ TuRC to the centriole lumen, impairs ciliogenesis. Consistent with this finding, recent work by Steib et al. (Steib et al., 2020) showed that POC5 depletion impairs centriole integrity. Moreover, we now show that in GCP4 mutant cells from patients, which display impaired ciliogenesis, the luminal signal of GCP4 is reduced, supporting the notion that luminal γ TuRC is required for ciliogenesis. Thus, components immediately upstream (POC5) and downstream (GCP4/ γ TuRC) of augmin in the luminal recruitment pathway are required for ciliogenesis. Finally, we can now show that loss of luminal augmin impairs centriole integrity (see below), similar to what was described for POC5 depletion (Steib et al., 2020). This further supports a role of luminal augmin in ciliogenesis.

- All the experiments implicating Augmin in luminal recruitment were done with HAUS6 RNAi and KO cells. That data is rather convincing. However, the authors argue that Augmin is required. To substantiate this statement, the authors need to show that other Augmin subunits are implicated in this process as it remains possible, with the data shown, that only HAUS6 is implicated. Consistent with this notion, HAUS6 is the NEDD1 binding component of Augmin, which in turn recruits gTuRCs and it thus remains possible, that the luminal function of Augmin is carried out by HAUS6 alone.

We would like to remind the reviewer that our previous manuscript already contained data showing luminal recruitment of other augmin subunits (endogenous HAUS5 and exogenously expressed HAUS8; Suppl. Fig. 1a) supporting the notion that augmin rather than HAUS6 alone is present in the lumen. As additional support, we now show that depletion of HAUS5, reduced both luminal HAUS6 and γ -tubulin (Fig. 2e).

- Do the authors have any evidence indicating that centriolar POC5, gTuRC and Augmin subunits (not only HAUS6) form a complex in vivo at endogenous levels? The BioID data and overexpression (filaments) data alone can entirely support that claim while biochemical evidence could.

While the reviewer does not specify what kind of "biochemical evidence" is requested here, there is already evidence in the literature that augmin and γ TuRC interact in interphase cells. For example, in a previous study we co-purified augmin subunits with γ TuRC from human interphase cells (Teixidó-Travesa et al., 2010). This study did not identify POC5 as γ TuRC interactor and we were also unable to observe this interaction by

immunoprecipitation of endogenous proteins from cell extract (our unpublished data). This is not unexpected though. In fact, many interactions are likely taking place specifically at centrosomes and therefore cannot be detected by conventional pulldown or immunoprecipitation from the soluble cell fraction. For this reason, BioID has been so successful in dissecting many of these interactions, e.g. (Gupta et al., 2015). We found this also to be the case for interactions between POC5, augmin and γ TuRC. As explained in response to reviewer #4, we have repeated the BioID experiment with whole cells and a more appropriate control sample. This confirmed POC5 as proximity interactor of HAUS6 and led to the identification of additional interactors including SAS6, a protein of the proximal centriole lumen (Fig. 3a).

Minor issues:

- CEP192 depletion does not effect NEDD1 luminal localization. The authors should confirm that POC5, HAUS6 and γ TuRC luminal localization is also unaffected as this would be consistent with their model but is not shown.

We have performed these analyses and confirmed that luminal POC5, HAUS6, and γ -tubulin are not affected by CEP192 depletion (Suppl. Fig. 2a,b).

- Raw BioID mass spec data should be provided in full.

The complete mass spec data is now included as a supplementary Excel file.

Reviewer #2 (Remarks to the Author):

In this paper, Schweizer et al. identify a non-canonical role for Augmin- γ -TuRC in maintaining centriole integrity to allow efficient ciliogenesis. They show that during centriole duplication, γ -TuRCs are recruited first to the outer centriole wall via CEP192 and later to the centriolar lumen via Augmin and Poc5. Both sites of γ -TuRC localisation require the γ -TuRC tethering factor NEDD1, which has been previously shown to mediate Augmin-dependent recruitment of γ -TuRCs to the sides of pre-existing microtubules and to mediate γ -TuRC recruitment via CEP192 to the PCM during mitosis. The authors used BioID to identify Poc5 as an Augmin interactor and then showed that depletion of Poc5 leads to a loss of Augmin- γ -TuRC within the centriole lumen. To assess a role for luminal Augmin- γ -TuRC the authors depleted Poc5 and showed that fewer centrioles remain after an extended mitotic arrest, suggesting that centriole integrity is impaired after loss of luminal Poc5 and Augmin- γ -TuRC. The authors also showed that the ability to form cilia is impaired after Poc5 depletion, suggesting that centriole integrity defects caused by the loss of luminal Augmin- γ -TuRC perturb cilia formation. They also observed cilia defects in a patient with a mutation in GCP4 (a γ -TuRC protein), suggesting that the cilia defects in this patient may be attributed to decreased centriole stability caused by a reduction in γ -TuRCs within the centriolar lumen.

This is an interesting paper that expands the range of γ -TuRC function and provides

insights into the γ -TuRC-related cause of human disease. However, I feel that prior to publication some links between observations and conclusions need to be supported by further data and I outline this below.

Major comments

1) At present, the authors conclude that centriole integrity is affected after the loss of Augmin- γ -TuRC from the centriole lumen and that this leads to the observed reduction in cilia / cilia defects, yet the authors lack direct evidence for this. Can the authors use expansion microscopy to identify centriole integrity defects (prior to loss of centrin that occurs only after a prolonged mitotic arrest) after Poc5 or HAUS6 depletion, perhaps by using acetylated tubulin and centrin antibody staining? Centriole width, or the distance between the centrin and the centriole wall, could increase, for example. Can the authors assess the structure of centrioles/basal within Poc5 depleted cells that have failed to generate cilia or that have shorter cilia?

These are all good suggestions. During revision of this manuscript, Steib et al. (Steib et al., 2020) showed centriole structure defects for POC5 RNAi cells using ultrastructure expansion microscopy (U-ExM). This directly supports the ciliogenesis defects that we described for POC5 RNAi cells. We then attempted to show similar defects for augmin-depleted cells. For this we tried RNAi of various augmin subunits, inducible HAUS8 KO cells (McKinley and Cheeseman, 2017), and inducible HAUS6 KO cells, which we newly generated during the revisions. While all of these approaches allowed efficient depletion in whole cells by western blot and in the mitotic spindle by IF, none efficiently removed augmin from the centriole lumen. To achieve this, we had to combine the knockdown or knockout of augmin with aurora kinase inhibitor treatment. This allowed cells with defective spindles after augmin depletion to pass through mitosis in the absence of cell division. Eventually these cells accumulated centrioles that lacked luminal augmin. We then employed U-ExM to analyse centrioles in these cells. Strikingly, centrioles in HAUS6 RNAi cells were shorter, some with severely reduced length. In HAUS6 knockout cells we also observed shorter centrioles, frequently with broken walls. This data (Fig. 5) strongly supports our conclusion that luminal augmin- γ TuRC promotes centriole integrity.

2) The authors link the patient mutation in GCP4 to cilia defects, hypothesising that these cilia defects are caused by centriole integrity defects. Can the authors recapitulate the cilia defects observed in GCP4 mutant patient fibroblasts by depleting GCP4 from RPE1 cells? The model would suggest that centriole integrity is also be perturbed in the extended mitosis assay after GCP4 depletion – can the authors show this?

We have performed both of these analyses. Indeed, GCP4 RNAi recapitulates the ciliogenesis defects observed in patient fibroblasts (Suppl. Fig. 5). Even though this was not specifically requested, we also added new data showing loss of luminal GCP4 signal in the patient cells by U-ExM (Fig. 6f,h). Moreover, we can now show that GCP4 knockdown also resulted in reduced centriole stability in the extended mitosis assay (Fig. 4f,g).

3) This relates somewhat to point 2. Is it Augmin or γ -TuRC, or both, that is necessary for centriole integrity? The authors suggest that the Y and L shaped structures (from EM images) previously observed to link centriole wall microtubules may represent Augmin, while ring structures in the lumen may represent γ -TuRCs. This is hard to reconcile with a model where Augmin binds and recruits γ -TuRCs to the lumen, as in this model one would expect Augmin and γ -TuRCs to be in very close proximity to each other.

Regarding the proximity of augmin and γ TuRC in the lumen, one needs to consider the size and orientation of both complexes. Augmin should be close to the inner wall where POC5 is located, and also be in contact with γ TuRC. However, γ TuRC has a diameter of ~ 29 nm and a height of ~ 27 nm (Zimmermann et al., 2020) and therefore part of it may protrude into the centriole lumen, well separated from augmin. Roughly, and depending on the detected subunits, one may expect POC5 to be closest to the inner centriole wall, γ -tubulin to be most luminal, and HAUS6 to occupy an intermediate position. Indeed, using U-ExM, we are now able to observe this differential localization of POC5, HAUS6, and γ -tubulin within the centriole lumen (Fig. 3c).

The EM observations may also suggest that Augmin is more important than γ -TuRC for centriole integrity. Depleting γ -TuRC components and assessing centriole integrity using the extended mitosis assay could help answer this, especially as this provides a reliable quantifiable readout of centriole integrity. It would then be interesting to show whether Augmin is still present after GCP4 or NEDD1 (i.e. γ -TuRC) depletion.

The extended mitosis assay to assess centriole stability after GCP4 depletion was added to the manuscript as described above (see point 2). We also added the requested data showing that HAUS6 is still present in the lumen after depletion of NEDD1 (Suppl. Fig. 2d).

Minor comments

4) Microtubule nucleation assay in Figure 1 – please point out that this is during interphase
This is stated in the legend.

5) Figure 2: the images of localisation defects after RNAi knockdowns would benefit from some form of quantification – either fluorescence intensity measurements, or fraction of cells without staining.

We have provided intensity measurements in Fig. 2h, i.

Reviewer #3 (Remarks to the Author):

Sub-centrosomal mapping identifies augmin- γ TuRC as part of a centriole-stabilizing scaffold by Schweizer et al. is an interesting paper presenting results mostly based around expansion microscopy of labelled centrioles under a number of conditions. They use the resultant data to draw significant new conclusions on centriole biology an vital cellular component involved in many biological processes.

Overall I think the paper is interesting and has useful new results for the field however I have significant concerns that several of the statistical results presented in the paper are overstated. Figure 1g presents values of: HAUS6 (absent): 0.5 ± 0.2 μ m, HAUS6 (lumen): 0.8 ± 0.2 μ m and states: p (HAUS6 absent/lumen) < 0.01 Yet if the absent value is 1 SD lower (chance $\sim 15\%$) and the lumen value is 1 SD higher (chance $\sim 15\%$) then the difference will be reversed, this has a chance of roughly $.15 \cdot .15$ or > 0.02 .

Several other comparisons have suspiciously high significance given the means and SD's quoted. These include Fig 4e, 24h data, Fig 5 e and Fig S2c. This last one is particularly strange, a percentage of intensity, which cannot be below 0, has a value of $100 \pm 50\%$ and yet a significance of < 0.001 . I don't believe that this data can be normally distributed and that significance is just mad.

We have re-evaluated all quantitative data and statistical analyses. In several cases we have added additional replicates and for all quantifications we have consulted our biostatistics unit to apply appropriate statistical testing, also considering the distribution of the data. We have also converted simple bar graphs to more descriptive plots. These changes did not affect our conclusions but strengthened and improved the presentation of the data.

Other more minor issues:

The paper relies extensively on the data from Expansion microscopy and yet the expansion factor ($\sim 4x$) is only cited in the methods section, and evidence for its validity is completely absent. The centriole is a well studied structure and simple measurements from their existing data measuring, for instance, the diameter of the microtubule structures could be compared to existing literature in optical microscopy and EM to give a real value. Various factors including pre-fixation, type of fixative, and gel recipe were found to affect expansion of centrioles relative to the gel (Gambarotto et al., 2019; Gambarotto et al., 2021; Sahabandu et al., 2019). Pre-fixation in methanol (as used for ExM in our study) may not achieve full isotropic expansion in all protocols, but has the great advantage that established immunostaining procedures and antibodies can be used and samples can be stored for later analysis. U-ExM in the absence of prior fixation, for example, shows more complete and isotropic expansion of centrioles, but requires immediate sample processing and has limitations in terms of availability of sufficient amounts of suitable antibodies (Gambarotto et al., 2019; Gambarotto et al., 2021). We have now quantified centriole diameters and lengths in expanded U2OS cells stained with acetylated tubulin antibodies (which yielded the best staining but may slightly underestimate the true dimensions) and processed by ExM (Tillberg et al., 2016). By dividing these values by the respective gel expansion factor we obtained an average diameter of ~ 190 nm and average length of ~ 355 nm. These values are similar to values obtained for centrioles in U2OS cells processed by U-ExM and polyE-tubulin staining (~ 192 nm and ~ 355 nm) (Gambarotto et al., 2021). We have included these data in the relevant method section.

I feel the introduction could do with more of a summary of existing state of the field regarding gamma-TuRC and its components and interactors as this information is not gathered together anywhere but scattered throughout the various results sections.

We have included information on γ TuRC composition and interactors in the introduction.

Fig 1g is reliant on very few data points, one experiment and 9-13 measurements per condition. This experiment needs to be repeated to generate more data to sustain the claimed significance, also see the point about the statistics above.

We have repeated the experiment several times and generated additional data for more accurate quantification.

The description of the imaging is incomplete, it needs information about the light sources and filters used.

This information was added to the method section.

The image processing section says "Images were processed in ImageJ and Photoshop (Adobe) and represent maximum projections of a deconvolved stack or a single section,

respectively" I don't understand what the respectively refers to and how was photoshop used to process the images?

We have updated the description in the method section to explain this better.

Several pieces in the methods refer to centrifugation speeds in rpm, these should be given as g values in order for people to be able to reproduce the methods.

We have included g values in all cases.

In the Mass spec methods section the authors say "Samples were cleaned through C18 tips (polyLC C18) and peptides were eluted with 80% acetonitrile/1% formic acid and consequently diluted to 20% acetonitrile/0.1% formic acid before loading into strong cation exchange columns (SCX)"

How do they dilute from 80:1 to 20:0.1, the ratios of the concentrations are totally different. "Samples were dried and reconstituted in H₂O with 3% acetonitrile/1% formic acid (1:8) in a total volume of 50 μ l." how does 3:1% relate to the 1:8 in brackets?

We have corrected and improved the description.

I was particularly impressed with the cartoons associated with every centriole image which made it much easier to understand the orientation and context of each image.

Thank you, that was our goal!

Reviewer #4 (Remarks to the Author):

This manuscript by Schweizer et al. describes the use of expansion microscopy to map the different populations of centrosomal γ TuRC, identify the augmin-mediated localization to the centriole lumen by interaction with POC5, and demonstrates that disruption of this luminal localization leads to centriolar and ciliary defects. I have focused my comments on the application of BioID to HAUS6 to identify interacting proteins in the centriole.

Specific comments:

This is a slightly unconventional approach to BioID that first biochemically separates the centrosomal material from the rest of the cell prior to biotin-affinity pulldowns. There is nothing wrong with this approach other than likely excluding associations that occur at sites distal to the centrosomes and/or the loss of proteins that are less strongly associated with the centrosome. Perhaps some language could be included about how this approach would exclude such associations. Western blots showing biotinylation of whole cells vs isolated centrioles could also help clarify what is being analyzed.

There appears to be a lack of a control that is typically used for BioID experiments. Most often this is the expression of the ligase that is not fused to anything, or is fused to a small targeting motif to force it to a specific subcellular compartment (e.g. a NLS for the nucleus). The purpose of this control is to exclude the many false positive associations that can occur due to affinity between various cellular proteins and the ligase. If the BioID-fused to the protein of interest detects a protein uniquely or at least substantially more than it is detected by the ligase-alone, then the association is considered valid. In this case a cell line not expressing any BioID was used as a control, which is a good control for non-specific associations with the streptavidin-coupled matrix or naturally biotinylated proteins but not for proteins with variable affinity to the BioID ligase of which there are many. Since

there is essentially no reporting of considerable numbers of other proteins associated with the bait and the functional association of HUS6 and POC5 are validated with other approaches it may not be necessary to repeat these BioID experiments with the proper controls; however, the limitations of the BioID experiment as performed should be noted. To address the concerns of this reviewer regarding appropriate controls for BioID and the use of isolated centrioles, we have repeated the experiment using BirA alone as a more appropriate control, as suggested, and performed the analysis with whole cells, rather than isolated centrioles. We identified overall more proteins, as expected. Among the top hits were multiple augmin and γ TuRC subunits, indicating that our approach was able to identify proteins proximal to the bait HAUS6. Importantly, POC5 was again among the hits, confirming the data initially obtain with isolated centrosomes.

- Berbari, N. F., Bishop, G. A., Askwith, C. C., Lewis, J. S. and Mykytyn, K.** (2007). Hippocampal neurons possess primary cilia in culture. *Journal of neuroscience research* **85**, 1095–1100.
- Gambarotto, D., Zwettler, F. U., Le Guennec, M., Schmidt-Cernohorska, M., Fortun, D., Borgers, S., Heine, J., Schloetel, J.-G., Reuss, M., Unser, M., et al.** (2019). Imaging cellular ultrastructures using expansion microscopy (U-ExM). *Nature Methods* **16**, 71–74.
- Gambarotto, D., Hamel, V. and Guichard, P.** (2021). Ultrastructure expansion microscopy (U-ExM). In *Methods in Cell Biology*, pp. 57–81. Elsevier.
- Gupta, G. D., Coyaud, É., Gonçalves, J., Mojarad, B. A., Liu, Y., Wu, Q., Gheiratmand, L., Comartin, D., Tkach, J. M., Cheung, S. W. T., et al.** (2015). A Dynamic Protein Interaction Landscape of the Human Centrosome-Cilium Interface. *Cell* **163**, 1484–1499.
- McKinley, K. L. and Cheeseman, I. M.** (2017). Large-Scale Analysis of CRISPR/Cas9 Cell-Cycle Knockouts Reveals the Diversity of p53-Dependent Responses to Cell-Cycle Defects. *Developmental Cell* **40**, 405-420.e2.
- Miki, D., Kobayashi, Y., Okada, T., Miyamoto, T., Takei, N., Sekino, Y., Koganezawa, N., Shirao, T. and Saito, Y.** (2019). Characterization of Functional Primary Cilia in Human Induced Pluripotent Stem Cell-Derived Neurons. *Neurochemical research* **44**, 1736–1744.
- Sahabandu, N., Kong, D., Magidson, V., Nanjundappa, R., Sullenberger, C., Mahjoub, M. R. and Loncarek, J.** (2019). Expansion microscopy for the analysis of centrioles and cilia. *J Microsc* **276**, 145–159.
- Steib, E., Laporte, M. H., Gambarotto, D., Olieric, N., Zheng, C., Borgers, S., Olieric, V., Le Guennec, M., Koll, F., Tassin, A.-M., et al.** (2020). WDR90 is a centriolar microtubule wall protein important for centriole architecture integrity. *eLife* **9**, e57205.

- Teixidó-Travesa, N., Villén, J., Lacasa, C., Bertran, M. T., Archinti, M., Gygi, S. P., Caelles, C., Roig, J. and Lüders, J.** (2010). The gammaTuRC revisited: a comparative analysis of interphase and mitotic human gammaTuRC redefines the set of core components and identifies the novel subunit GCP8. *Molecular biology of the cell* **21**, 3963–3972.
- Tillberg, P. W., Chen, F., Piatkevich, K. D., Zhao, Y., Yu, C.-C. J., English, B. P., Gao, L., Martorell, A., Suk, H.-J., Yoshida, F., et al.** (2016). Protein-retention expansion microscopy of cells and tissues labeled using standard fluorescent proteins and antibodies. *Nat Biotechnol* **34**, 987–992.
- Viais, R., Watanabe, S., Villamor, M., Palenzuela, L., Lacasa, C., Fariña, M. and Lüders, J.** (2020). Disruption of augmin-mediated microtubule nucleation in neural stem cells causes p53-dependent apoptosis and aborts brain development. *bioRxiv* 2020.11.18.388694.
- Zimmermann, F., Serna, M., Ezquerra, A., Fernandez-Leiro, R., Llorca, O. and Luders, J.** (2020). Assembly of the asymmetric human γ -tubulin ring complex by RUVBL1-RUVBL2 AAA ATPase. *Science Advances* **6**, eabe0894.

REVIEWERS' COMMENTS

Reviewer #1 (Remarks to the Author):

The authors did a good job addressing my concerns. I understand the technical issues they are experiencing. I would recommend they discuss this a bit better in the text.

Reviewer #2 (Remarks to the Author):

the authors have addressed all my concerns. The paper is ready for publication

Reviewer #3 (Remarks to the Author):

Sub-centrosomal mapping identifies augmin-gTuRC as part of a centriole-stabilizing scaffold by Schweizer et al..

The authors have worked to improve the paper and answered most of the question I put forward in my original review. There is one major new issue introduced in this version.

Although the authors explain their statistical analysis in much fuller detail they have removed basically all the numerical data from the paper. Data which was previously shown as graphs but also presented as more traditional means \pm SD values is now only shown in graphs, leaving the reader to estimate the numeric values. This dramatically reduces the usefulness of the results presented and makes replication effectively impossible. I think this paper is unpublishable without these numeric values.

Minor issues:

I don't think the authors understood my original point about their expansion factor. My expectation is the the expansion factor is the largest variable, and that the known microtubule ring size can be used to calibrate how much expansion has occurred. This is not a critical factor but would be an interesting piece of data to have.

If the authors reintroduce the numerical data they have removed the paper is of a publishable standard.

Reviewer #4 (Remarks to the Author):

The authors have addressed my concerns with the BioID experiments. It is worth noting that the #4 hit ACACB is a naturally biotinylated mitochondrial protein that is likely background that was detected as more abundant than in the BioID-only samples due to either chance or more likely that the ligase-alone generally biotinylates more proteins and thus better outcompetes the endogenously biotinylated proteins during affinity capture than occurs in samples that have less biotinylated proteins. This could be reasonably excluded as a a potential interacting/proximate protein for the reason noted above or at least the high likelihood of it being background should be noted.

REVIEWERS' COMMENTS

Reviewer #1 (Remarks to the Author):

The authors did a good job addressing my concerns. I understand the technical issues they are experiencing. I would recommend they discuss this a bit better in the text.

We have added text modifications to the results section to make this clearer.

Reviewer #2 (Remarks to the Author):

the authors have addressed all my concerns. The paper is ready for publication

Reviewer #3 (Remarks to the Author):

Sub-centrosomal mapping identifies augmin-gTuRC as part of a centriole-stabilizing scaffold by Schweizer et al..

The authors have worked to improve the paper and answered most of the question I put forward in my original review. There is one major new issue introduced in this version.

Although the authors explain their statistical analysis in much fuller detail they have removed basically all the numerical data from the paper. Data which was previously shown as graphs but also presented as more traditional means+-SD values is now only shown in graphs, leaving the reader to estimate the numeric values. This dramatically reduces the usefulness of the results presented and makes replication effectively impossible. I think this paper is unpublishable without these numeric values.

We have included the requested values in the source data files.

Minor issues:

I don't think the authors understood my original point about their expansion factor. My expectation is the the expansion factor is the largest variable, and that the known microtubule ring size can be used to calibrate how much expansion has occurred. This is not a critical factor but would be an interesting piece of data to have.

As explained in our previous rebuttal, depending on the protocol used, expansion of centrioles is not a good measure of the true expansion factor, since centrioles are

known to expand non-isotropically in some protocols. Rather than determining expansion indirectly, estimating the expansion factor directly by measuring gel size, as done in our manuscript, is the approach that has been used in all reference expansion microscopy method papers. We also note that we are not deriving any absolute measurements from our expansion microscopy, but only comparative data, which are unaffected by the precise expansion factor.

If the authors reintroduce the numerical data they have removed the paper is of a publishable standard.

Reviewer #4 (Remarks to the Author):

The authors have addressed my concerns with the BioID experiments. It is worth noting that the #4 hit ACACB is a naturally biotinylated mitochondrial protein that is likely background that was detected as more abundant than in the BioID-only samples due to either chance or more likely that the ligase-alone generally biotinylates more proteins and thus better outcompetes the endogenously biotinylated proteins during affinity capture than occurs in samples that have less biotinylated proteins. This could be reasonably excluded as a a potential interacting/proximate protein for the reason noted above or at least the high likelihood of it being background should be noted.

We thank the reviewer for detecting this. We have removed the protein from the lists in the figure (not from the supplementary raw data file) and added a note to the legend, explaining that this protein is likely background.